

# Improving the representation of major Indian crops in the Community Land Model version 5.0 (CLM5) using site-scale crop data

K. Narender Reddy[1], Somnath Baidya Roy[1], Sam S. Rabin[2], Danica L. Lombardozzi[2], Gudimetla
Venkateswara Varma[1], Ruchira Biswas[1], and Devavat Chiru Naik[3]

[1]Centre for Atmospheric Sciences, Indian Institute of Technology Delhi, New Delhi, 110016, India
[2]Climate and Global Dynamics Laboratory, National Centre for Atmospheric Research, Boulder, 80307, USA
[3]Department of Civil Engineering, Indian Institute of Technology Delhi, New Delhi, 110016, India

*Correspondence to*: K. Narender Reddy (knreddy@cas.iitd.ac.in)

**Abstract.** Accurate representation of croplands is essential for simulating terrestrial water, energy, and carbon fluxes over India because croplands constitute more than 50% of the Indian land mass. Spring wheat and rice are the two major crops grown in India, covering more than 80% of the agricultural land. The Community Land Model version 5 (CLM5) has significant errors in simulating the crop phenology, yield, and growing season lengths due to errors in the parameterizations of the crop module, leading to errors in carbon, water, and energy fluxes over these croplands. Our study aimed to improve the 15 representation of these crops in CLM5. Unfortunately, the crop data necessary to calibrate and evaluate the models over the Indian region is not readily available. In this study, we used a comprehensive spring wheat and rice database that is the first of its kind for India which is created by digitizing historical observations. We used eight spring wheat sites and eight rice sites and many of the sites have multiple growing seasons, taking the tally up to nearly 20 growing seasons for each crop. We used this data to calibrate and improve the representation of the sowing dates, growing season, growth parameters, and base 20 temperature in the CLM5 model. The modified CLM5 performed much better than the default model in simulating the crop phenology, yield, carbon, water, and energy fluxes when compared with the site scale data and remote sensing observations. For instance, Pearson's $r$ for monthly LAI improved from 0.35 to 0.92, and monthly GPP improved from -0.46 to 0.79 compared to MODIS monthly data. The $r$ value of the monthly sensible and latent heat fluxes improved from 0.76 and 0.52 to 0.9 and 0.88, respectively. Moreover, because of the corrected representation of the growing seasons, the seasonality of the 25 simulated irrigation now matched the observations. This study demonstrates that the global land models must use region-specific parameters rather than global parameters for accurately simulating vegetation processes and, eventually, land surface processes. Such improved land models will be a great asset in investigating the global and regional scale land-atmosphere interactions and developing future climate scenarios.

## 1 Introduction

Land Surface Models (LSMs), the land components of Earth System Models (ESMs), represent a wide variety of processes, including energy partitioning, carbon and mass exchange, and interaction with the hydrological cycle, to name a few. LSMs



provide boundary conditions and interact with various components of ESMs (Fisher and Koven, 2020; Strebel et al., 2022). LSMs have come a long way, from a very basic representation of energy budget at the surface level to a very complex state where each grid cell consists of multiple land units and unique interaction of the individual land unit with the atmospheric

forcings (Blyth et al., 2021). LSMs use sophisticated parametrization and modules to represent the complex land surfaces and their interactions with other components of ESMs. One important component of LSMs that significantly impacts not only land processes, but also atmospheric processes is the agricultural land. LSMs strive towards a realistic depiction of agricultural land cover and its processes. Until the last decade, the depiction of crops was mostly constrained to rudimentary models that do not include agricultural practices such as irrigation and fertilization or simply depicted crops as natural grassland (Elliott et al.,

2015; McDermid et al., 2017). Enhancements to crop modules gave LSMs a greater capacity to investigate changes in water and energy cycles from croplands, as well as crop yield, in response to climate, environment, land use, and land management variations. Recent studies provide valuable insights for enhancing the accuracy of simulating biogeophysical and biogeochemical processes at both regional and global scales in LSMs (Lokupitiya et al., 2009; Lobell et al., 2011; Levis et al., 2012; Osborne et al., 2015; McDermid et al., 2017; Lawrence et al., 2018; Lombardozzi et al., 2020; Boas et al., 2021).

The Community Land Model (CLM) has, since version 4.0, included a prognostic crop module based on the Agro-Ecosystem Integrated Biosphere Simulator (Agro-IBIS) (Levis et al., 2012; Lawrence et al., 2018; Lawrence et al., 2019). This module can simulate the soil-vegetation-atmosphere system, including crop yields. The most recent version of CLM, known as CLM5, is a leading land surface model with an interactive crop module representing crop management. The module comprises eight crop types that are actively managed: temperate soybean, tropical soybean, temperate corn, tropical corn, spring wheat, cotton,

rice, and sugarcane. It also contains irrigated, non-irrigated, and unmanaged crops (Lombardozzi et al., 2020). Currently, CLM5 is the sole land surface model incorporating dynamic spatial patterns of significant crop varieties and their management (Lombardozzi et al., 2020). Although CLM5 showed advancements compared to its previous versions, limited research conducted at the point and regional scales indicates that it may provide poor phenology and yield predictions for certain crops (Chen et al., 2018; Sheng et al., 2018; Boas et al., 2021). The energy and carbon fluxes are highly affected by inaccuracies in

crop phenology, particularly concerning the timing of planting and harvesting.

The Indian subcontinent is a very important landmass that significantly affects the earth's system energy, water, and carbon fluxes. Nearly 50% of the land cover is used for agriculture in India, and two major cereal crops, wheat, and rice, occupy nearly 80% of the total agricultural land. However, CLM5 simulations of rice and wheat over the Indian subcontinent show large biases (Lombardozzi et al., 2020). The irrigation patterns simulated by CLM have a bias in seasonality, which is highlighted

by Mathur and AchuthaRao. (2019). Irrigation is an essential feature of the croplands in India, especially during the rabi season (Gahlot et al., 2020) for wheat and in dry regions for rice. Therefore, the bias in irrigation points to the lack of accurate representation of Indian crops.

Gahlot et al. (2020) used an LSM (Integrated Science Assessment Model; ISAM) to investigate the wheat croplands of India. The major drawback of the study was the lack of enough site-scale observations to calibrate and validate the model while

covering the broad growing conditions of India. Therefore, in this study, we aim to investigate and improve the representation





of major Indian crops—wheat and rice—in the latest version of CLM (CLM5.0). We used site-scale observations from multiple sites to calibrate the parameters essential for the crop module in CLM5 and evaluate the model. The site-scale observations cover various climatic conditions experienced by crops in India, thus making this a robust calibration of an LSM. Further, we aimed to quantify the impacts of realistic representation of Indian crops on various land processes such as irrigation, gross

primary production, latent heat, and sensible heat.

The current paper is structured as follows: First, we briefly describe the CLM5 model, and the site scale data used in this study. Then, we describe the shortcomings of CLM5 in simulating Indian crops, comparing them to the observations. Next, we dive into the need for modifications in CLM5 and the changes made to parameters and the source code of the CLM5. The results section compares our improved model at site and regional scales. We compare the CLM5 simulations against observed Leaf

Area Index (LAI), yield, and growing season length at site scale. At the regional scale, we compare against yield, irrigation patterns, LAI, Gross Primary Production (GPP), Latent Heat flux (LH), and Sensible Heat flux (SH) observations. Finally, we discuss the impact of the study and the conclusions.

## 2 Materials and Methods

### 2.1 Community Land Model version 5 (CLM5.0)

CLM5 is the latest version of the land component in the Community Earth System Model (CESM) (Lawrence et al., 2018, 2019). The biogeochemistry mode of CLM5 (CLM5-BGC) is widely used to estimate the water, energy, and carbon fluxes in various climatic zones (Cheng et al., 2021; Denager et al., 2023; Song et al., 2020; Seo and Kim, 2023). The biogeochemistry and crop module of CLM5 (BGC-Crop) is modified in various studies to meet regional constraints, and the resulting impact on various fluxes is analyzed (Boas et al., 2021; Raczka et al., 2021; Boas et al., 2023; Yin et al., 2023). Studies show that

incorporating agriculturally managed land cover can improve the general representation of biogeochemical processes (Boas et al., 2021). The CLM5 crop module includes new crop functional types, updated fertilization rates and irrigation triggers, a transient crop management option, and some adjustments to phenological parameters (Lombardozzi et al., 2020).

CLM5 has a better representation of the land surface by using a tile representation. This allows the model to have various land types inside a grid cell. In its latest version, the model supports 79 plant functional types with 32 rainfed and 32 irrigated crop

types. The complex representation of the land surface makes CLM5 a better model on various metrics tested by International Land Model Benchmarking (ILAMB) (Collier et al., 2018).

The current study used the CLM5 model in the data atmosphere mode i.e., not interacting with the atmosphere. The GSWP3 atmospheric data is used for the simulations. We ran CLM5 at two different spatial resolutions from 2000 to 2014: site-scale simulations to calibrate the crop module and regional simulations to evaluate the calibrated model against remote sensing data

and derived surface flux data (Sect. 2.5). The default CLM5 model is referred to as CLM5_Def throughout this paper. CLM5_Mod1 and CLM5_Mod2 are the two setups of the model developed in this study, and they are described in detail in Section 2.3.

### 2.1.1 Site scale simulations





For site scale simulations, we created domain, surface, and land use time series data for the respective sites (for details on sites,

Section 2.2, and Figure 1). The resolution of the data is 0.1° and has one grid cell with the site at its center. The method used to generate the data is available in the documentation of Reddy et al., (2024). The domain file represents the spatial extent of our simulation. The surface data represents the local soil and surface properties. The land use time series reflects the varying land-use land cover change from 1850 to 2015 at sites. Spin-up at each site is carried out for 200 years in accelerated deposition mode (AD mode) and 400 years in normal mode. The GSWP3 atmospheric data is used for the site scale simulations.

**2.1.2 Regional-scale simulations**

For regional scale simulations, we fixed the domain between 60 °E to 100 °E and 0 °N to 40 °N (Figure S1), covering the Indian subcontinent. The domain, surface, and land use time series data are generated for the domain mentioned above with a spatial resolution of 0.5° (files available at Reddy et al., 2024). The spin-up for the regional case is carried out in two stages. Two hundred years of spin-up in AD mode and 400 years in normal mode. The simulation data at the end of 400 years is used

as initial conditions for our regional simulations. The regional simulations are run from 1995 to 2014, and the data from 2000 to 2014 is used for the analysis. The GSWP3 atmospheric data is used as atmospheric forcing for the regional scale simulations.

**2.2 Site scale crop data**

Site-scale data of the type and quality required for calibrating and validating crop models are not readily available in India. This is unfortunate because plenty of data has been collected, but they have never been properly archived. India has invested

heavily in agricultural studies and has built nearly 70 agricultural institutes nationwide since the green revolution in the 1960s, with each state having at least one institute dedicated to studying regional crops. Master's and PhD student theses from these institutes, many containing site-scale observations, were recently consolidated and brought into the public domain in the KrishiKosh repository (Veeranjaneyulu, 2014). However, the data is difficult to extract from these theses because of the data collection and reporting structure differences followed by various institutes. For this study, we assembled data on wheat and

rice in a formatted, machine-readable format that can be downloaded and used for model development. The data is available on the PANGEA repository (Varma et al., 2024). We used the site scale data (years 2000 to 2014) generated by Varma et al. (2024) to evaluate our CLM5 model (Table S1 and Figure 1).

**2.3 Improvements in CLM5**

The parameters impacting planting and growing stages in CLM5 are minimum and maximum planting dates, planting

temperature, planting temperature, base temperature for Growing Degree-Day (GDD) calculations, minimum GDD for crop emergence, and GDD threshold for crop grain fill. The minimum planting temperature and the average minimum planting temperature of the growing season govern the planting date of the crop in CLM5. The base temperature defines the crop growth rate and the accumulation of GDD. Crop growth has different phases: emergence, flowering, grain fill, and maturity. The CLM5 model simulates the crop growth phases using the accumulated GDD. Therefore, base temperature becomes a critical

parameter that defines the crop growth in CLM5. The base temperature and maximum GDD control the longevity of each phase in crop growth. The allocation to the grain starts once the crop reaches the grain fill stage, which is controlled through the "grnfill" parameter in CLM5. The "grnfill" parameter defines the threshold for initiating the grain-filling stage as a fraction





of the GDD required for maturity (hybgdd in Table 1). Growing season length in CLM5 is directly controlled through base temperature. The lower the base temperature, the faster the GDD accumulation and the shorter the growing season length.

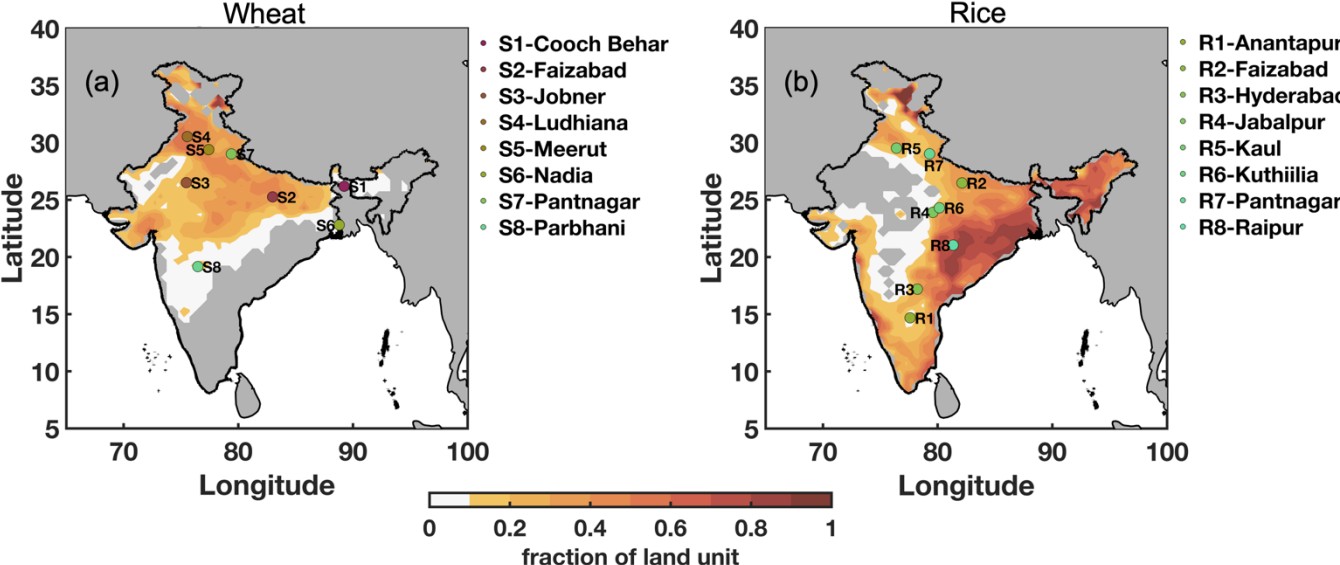


**Figure 1: Location of sites used in the current study for calibrating and validating the major Indian crops (a) spring wheat and (b) rice.**

The improvements to the wheat and rice crops in CLM5 are made in two steps. We first perform a literature survey and conduct sensitivity experiments to find the best-performing parameters shown in Table 1 (Section 2.3.1). The CLM5_Mod1 setup is

the result of the new parameter values. Second, we calibrate the latitudinal variation in base temperature through sensitivity experiments (Section 2.3.2). The CLM5_Mod2 setup results from calibrating the latitudinal variation in base temperature. Changes in the source code of CLM5 were necessary to facilitate the incorporation of changes made to parameters (see Section 2.3.1.1).

**2.3.1 Improvements in CLM5_Mod1**

**2.3.1.1 Spring Wheat**

CLM5_Def simulated the wheat growth from April to August. This is in stark contrast with ground reality, where Indian farmers sow spring wheat in late October to early November and harvest in late March or April (rabi season) (Sacks et al., 2010; Gahlot et al., 2020). To implement a realistic growing season, we performed sensitivity simulations by varying the planting window of 45 days, from mid-October to late November (see Table S2). The planting window shown in Table 1

produced the best results in lowering the bias in simulated LAI, yield, and growing season length and, therefore, is used in CLM5_Mod1. The CLM5_Def base temperature for spring wheat is 0 °C, but during our literature survey, we found the optimal base temperature for wheat in India is 5 °C (Mukherjee et al., 2019; Mehta and Dhaliwal, 2023). The planting temperature threshold in CLM5 for wheat is low compared to observations in India (Rao et al., 2015; Asseng et al., 2016;





Mukherjee et al., 2019). The grain fill threshold of 0.6 for wheat performed well amongst tested values in our sensitivity studies
(Table S2), and therefore, we did not change the parameter value.

**2.3.1.2 Rice**

CLM5_Def simulated rice growth from January to May. In contrast, rice is grown in India during the monsoon season due to the high water requirements of the rice crop. Rice is sown in the last week of June to early July and harvested at the end of October and early November, also known as the Kharif season. Many regions in India grow rice during the summer and rabi
seasons, which meet their water requirements mainly through irrigation. The rice crop area grown in summer and rabi is very low compared to the rice crop grown in the kharif season (Biemans et al., 2016). Therefore, we confined ourselves to the major rice growing season (kharif season) to calibrate the model. A sensitivity study is conducted with a planting window of 45 days, from early June to late July (Table S2). The planting window shown in Table 1 for rice gave the best results. The base temperature used for rice crop (10 °C) in CLM5_Def is the same as that observed in the literature for the Indian region (Thakur
et al., 2022). However, we found that the planting temperature observed in India differs from those used in CLM5_Def (Jat et al., 2019; Kumar et al., 2023). The grain fill threshold used for rice in the CLM5_Def case resulted in very poor LAI and yield simulations, which was earlier recognized by Lu and Yang (2021) while studying rice in China using CLM. Through a sensitivity test, we found that the grain fill threshold of 0.65 performed the best in simulating LAI and yield for rice amongst the tested grain fill values in Table S2.

The parameter of growing degree-days required for maturity (hybgdd) in both wheat and rice was performing well during our sensitivity simulations, and, therefore, its value is not altered. Table 1 shows all the parameters changed in the default CLM5 to improve wheat and rice crop growth for the Indian region.

**Table 1: Parameter values for spring wheat and rice in the CLM5 crop module**

| Parameter | Description (units) | Wheat | | Rice | |
|---|---|---|---|---|---|
| | | CLM5_Def | CLM5_Mod1 | CLM5_Def | CLM5_Mod1 |
| min_NH_planting_date | Minimum planting date for the Northern hemisphere (MMDD) | 401 | 1115 (calibrated in this study) | 101 | 701 (calibrated in this study) |
| max_NH_planting_date | Maximum planting date for the Northern hemisphere (MMDD) | 615 | 1231 (calibrated in this study) | 228 | 815 (calibrated in this study) |
| min_planting_temp | Average 5 day daily minimum temperature needed for planting (K) | 272.15 | 283.15 (Rao et al., 2015) | 283.15 | 294.15 (Kumar et al., 2023) |
| planting_temp | Average 10-day temperature needed for planting (K) | 280.15 | 290.15 (Asseng et al., 2016; Mukherjee et al., 2019) | 294.15 | 300.15 (Jat et al., 2019) |
| baset | Base Temperature (°C) | 0 | 5 (Mukherjee et al., 2019; Mehta and Dhaliwal, 2023) | 10 | 10 (Thakur et al., 2022) |
| grnfill | Grain fill parameter | 0.6 | 0.6 | 0.4 | 0.65 (calibrated in this study) |





| hybgdd | Growing Degree Days for maturity (°C-days) | 1700 | 1700 | 2100 | 2100 |
| baset_mapping | Switch to turn on/off the latitudinal variation in baset in tropics | 'constant' | 'constant' | 'constant' | 'constant' |

#### 2.3.1.1 Source code changes

Along with the parameter changes, we had to change the model source code to fix a bug with northern-hemisphere crop seasons that start in one calendar year and finish in the next. The code added to the module CNPhenologyMod.F90 starting at line 2001 is shown below. The code changes are available at Reddy et al. (2024).

```
Line no.
2001    if (cphase(p) == 4._r8) then
2002      if (idop(p) > jday) then
2003          cropplant(p) = .false.
2004          idop(p) = NOT_PLANTED
2005      else
2006          cropplant(p) = .true.
2007      end if
2008    else
2009    end if
```

where cphase = crop phenology phase, idop = day of planting, and jday = Julian day of the year.

This bug is fixed in more recent versions of CLM, starting with tag ctsm5.1.dev131. A bug was also fixed to make CLM use user-specified values of parameters latvary_intercept and latvary_slope, which allow latitudinal variation of base temperature. More recent versions of CLM, starting with tag ctsm5.1.dev155, include this fix.

#### 2.3.2 Mod2 case parameters: Varying base temperature by latitude

CLM5 can vary CFT base temperature by latitude to account for cultivars bred for optimal performance in different climates. Currently, only spring wheat and sugarcane have these capabilities turned on. We extended this latitudinal variability to rice and improved the existing one for spring wheat in India. The latitudinal variation in base temperature is defined by two parameters: latvary_intercept and latvary_slope. The equation in the model that uses these parameters is:

$$Tbase_{lat} = Tbase + latvary_{intercept} - \min\{latvary_{intercept}, latvray_{slope} * |latitude|\} \quad ......(1)$$

*latvary_slope* and *latvary_intercept* define the latitudinal extent of the base temperature variation. *Tbase* refers to the base temperature used for GDD calculation beyond the latitudinal limit.

We conducted sensitivity studies to find the optimal latvary_intercept and latvary_slope values for wheat and rice. We ran the site scale simulations at experimental sites with data on LAI, yield, and growing season length. This resulted in 14 sites in total (Table S1), 7 for rice and 8 for spring wheat. Bias is considered to calibrate the model. The bias formula used in the study is:

$$Mean\ Absolute\ Bias\ (MAB)\ =\ \frac{\sum|CLM_{var} - Obs_{var}|}{\sum(Obs_{var})} \quad ........(2)$$

where, *var* is LAI, yield, or growing season length.

MAB is calculated for LAI, yield, and growing season length. The overall bias, used as our evaluation metric during calibration, is calculated as the equally weighted average of mean absolute bias in LAI, yield, and growing season length.



We ran ten simulations at each site to test the sensitivity of base temperature on crop growth and evaluate optimal base temperatures. Two of the simulations CLM5_Def and CLM5_Mod1 use the parameter values showed in Table 1. The other

eight simulations at each site used the same parameter set as given in Table 1 but with a base temperature (based) changed relative to the CLM5_Mod1 values given thereby ± [1, 2, 3, 4] °C. The total number of site scale simulations conducted and used for this sensitivity analysis is 150 (15 sites, 10 simulations per site). These simulations helped us understand the bias in the CLM5_Def and CLM5_Mod1 simulations, and the sensitivity of base temperature on crop growth and phenology at individual sites.

Figure 2 represents the sensitivity of wheat and rice crop growth to base temperature in the site-scale sensitivity simulations. The y-axis depicts the overall bias in the model (sum of bias in LAI, yield, and the growing season length). In the case of wheat, the CLM5_Def parameterization has the highest bias at all sites in the range 0.45-0.8 (markers in dark green color in Figure 2(a)). The bias in CLM5_Mod1 is in the range of 0.1-0.3 (markers in light green in Figure 2(a)). The bias in sensitivity experiments with base temperature at each site is shown in Figure 2 with grey markers and the least bias at each site is shown

in black marker. The base temperature of 5 °C produced the least bias at three sites (Pantnagar, Meerut, and Jobner). The remaining four sites have the least bias at temperatures above 5 °C. Ludhiana site, which is above 30 °N, performed the best at 6 °C, while Parbhani, Cooch Behar, and Faizabad had the least bias at 7 °C. The three sites having the least bias at 7 °C are in the central and southern parts of the wheat-growing regions of India. The sites performing best at 5 °C are in the northern part of the wheat-growing region.

In the case of rice, CLM5_Def has the highest bias, ranging from 0.5-0.95 (shown in dark green markers in Figure 2(b)). The difference between the CLM5_Def and CLM5_Mod1 cases is the grain fill parameter (Table 1). Using 0.65 as grain fill drastically improved the rice crop simulations. The bias in CLM5_Mod1 is in the range of 0.1-0.3 (markers in light green in Figure 2(b)). All the sensitivity experiments used the grain fill parameter of 0.65. The sensitivity of base temperature in rice showed that the sites in the southern rice growing regions (lower than the Tropic of Cancer, latitude < 23.5 °N) have the least

bias at 11 or 12 °C. The sites in the central rice growing regions (23.5 °N < latitude < 29 °N) have the least bias while using base temperatures of 8 or 9 °C. Finally, the sites towards the country's northern parts (latitude > 29 °N) perform best at 9 °C as the base temperature. Therefore, not all sites perform optimally at a single base temperature, and a latitudinal variation in base temperature can improve the rice crop simulations.





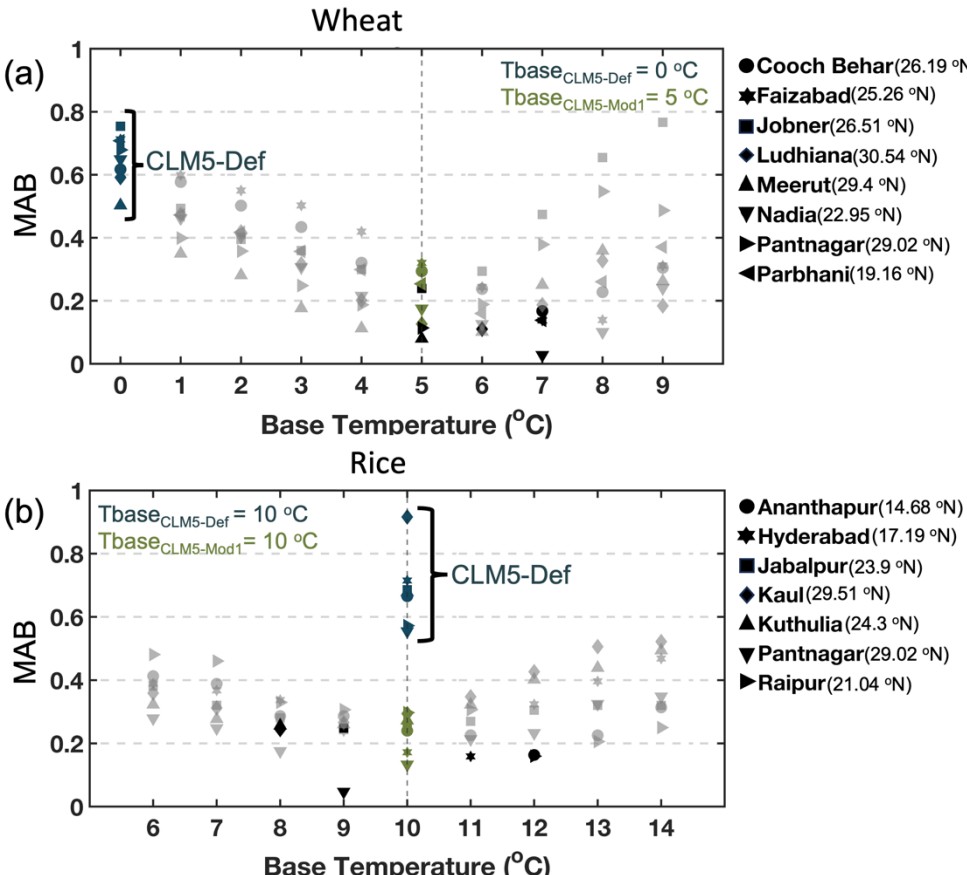

**Figure 2: The overall bias in the site scale simulations during the sensitivity study of base temperature (x-axis) for (a) spring wheat and (b) rice. The y-axis shows the overall bias (mean of absolute bias in LAI, yield, and growing season length). The dark green markers show the bias in the Def case at a site, the light green marker shows the bias in the Mod1 case at a site, and the black marker shows the lowest bias simulated at a site. The grey markers show the bias simulated in the sensitivity study of base temperature at a site. The legend shows the name and latitude of the sites.**

The base temperature at which the least bias is observed at each site and the corresponding latitude is noted for wheat and rice crops (Table S3). Using the ordinary least squares method, the values for latvary_intercept and latvary_slope are calculated which satisfy Eq. (1) for wheat and rice (Table 2 and Figure S2). Figure S2 shows the linear fit of the base temperature at which the lowest bias is observed (Table S3) and the latitude of the site. The linear fit has a high $R^2$ of 0.64 for wheat and 0.68 for rice.

The Mod2 version of the model used these parameters. In CLM5_Mod2 we used the baset_mapping equal "varytropicsbylat" in the CLM namelist to turn on the latitudinal variation in base temperature in the model. To incorporate the latitudinal variation for rice crops in CLM5, an addition to the code of CropType.F90 is made at line 602 (see supplementary material).

**Table 2: Latitudinal variation parameters for spring wheat and rice**

| Parameter name | Wheat | Rice |
|---|---|---|





| | CLM5_Def | CLM5_Mod2 | CLM5_Def | CLM5_Mod2 |
|---|---|---|---|---|
| baset | 0 | 5.4* | 10 | 9* |
| latvary_intercept | 12 | 6* | NA | 6.8* |
| latvary_slope | 0.4 | 0.19* | NA | 0.26* |

\* significant at p<0.05 using the t-statistic of the two-sided hypothesis test. NA – Not Applicable

## 2.4 Evaluation metrics

The comparison of CLM5 simulations with observations at site scale and regional scale used four evaluation parameters: mean absolute bias (MAB) (Eq. 2), root mean square error (RMSE), Pearson's $r$, and Kling-Gupta Efficiency (KGE, Gupta et al., 2009). MAB is the normalized deviation from the observations, with values close to 0 indicating good performance. RMSE is the mean deviation of model simulations from observations. Pearson's $r$ gives the correlation between the model estimates and observations. KGE (Eq. 3) offers a diagnostic insight into the model performance because it is a composite of correlation, bias, and variability.

$$KGE = 1 - \sqrt{(r-1)^2 + (\beta-1)^2 + (\gamma-1)^2} \quad \text{.... Eq. (3)}$$

$$\beta = \frac{\mu_{CLM}}{\mu_{Obs}}$$

$$\gamma = \frac{\sigma_{CLM}}{\sigma_{Obs}}$$

where KGE is the Kling-Gupta Efficiency, $r$ is the Pearson's coefficient between CLM simulated variable and observations, $\beta$ is the bias ratio (ratio of means-μ of the modeled and observation values) and $\gamma$ is the variability ratio (ratio of standard deviations- $\sigma$ of modeled and observation values). KGE, $r$, $\beta$, and $\gamma$ have their optimum at unity.

KGE is widely used in hydrological modeling because of its easy formulation and interpretation (Kling et al., 2012). KGE also makes sense from an agroecosystem point of view because we are interested in reproducing temporal dynamics, as well as preserving the spatial variation in crop growth caused by diverse climatic conditions in the Indian region, which are given by the first ($\beta$) and the second ($\gamma$) moments, respectively.

The Taylor diagram (Taylor, 2005) is used to assess the CLM5 model. The Taylor diagram summarizes the relative skill with which different models imitate the pattern in observations. The three versions of the CLM5 model from the study are represented by triangles on the Taylor diagram (Figure 4). The distance between each CLM5 setup and the point displayed as a black star (observation data) on the Taylor diagram indicates how accurately each model reproduces observations. Three statistics of the simulated fields are plotted on the Taylor diagram: a) the centered RMS error that is proportional to the distance from the point on the x-axis shown as a black star (dark green contours); b) the standard deviation that is proportional to the radial distance from the origin (grey semi-circular contours); and c) the Pearson correlation coefficient that is proportional to the azimuthal angle (light grey contours). Higher correlation, lower RMS error, and smaller standard deviation characterize the most accurate CLM5 configuration.




## 2.5 Model evaluation at the site scale

We compared the CLM5_Def, CLM5_Mod1, and CLM5_Mod2 simulations against the site-scale observations. We evaluated three crop variables, LAI, growing season length, and yield. We used four evaluation metrics MAB, RMSE, Pearson's *r*, and KGE (described in Section 2.4). Because the count of observation data points is low, we used the bootstrapping method to
estimate the significance of improvement from CLM5_Def to CLM5_Mod1 and CLM5_Mod1 to CLM5_Mod2. Bootstrapping is carried out with 10000 samples for each evaluation metric and the Student's T-test is conducted to check if each model improvement is performing significantly better ($p<.05$) than its predecessor. Table 3 shows the above-mentioned evaluation metrics. Note that 64% of the observations are used for calibration and the rest marked with "*" in Table S1 are used for validation.

## 2.6 Model evaluation at the regional scale

### 2.6.1 Yield

We compared the yield simulated by CLM5 against the EarthStat yield data (Ray et al., 2012) retrieved from the "Harvested Area and Yield for 4 Crops (1995–2005)" dataset. EarthStat yield data is available at a spatial resolution of 0.1°x0.1° and is given as a five-year average. In this study, we used the 2005 EarthStat data (representing the average yield from 2003 to 2007)
regridded to 0.5°x0.5° and compared it against the CLM5 simulated yield data averaged from 2003 to 2007.

### 2.6.2 Irrigation

An investigation on irrigation using a climate model in Indian croplands was carried out by Biemans et al. (2016). The study highlighted the necessity of improving the cropping patterns to improve the irrigation patterns. We compared the annual mean irrigation pattern simulated by three versions of CLM5 against the annual mean irrigation water demand for spring wheat and
rice from Biemans et al. (2016). The irrigation pattern data from Biemans et al. (2016) was not available as a supplement. Therefore, we extracted data from the Figure 5 of Biemans et al. (2016).

### 2.6.3 LAI and GPP

We compared the regional scale model simulations against the Moderate Resolution Imaging Spectroradiometer (MODIS) 8-day GPP (MOD17A2HV006) (Running and Zhao, 2015) and LAI (MOD15A2HV0061) (Myneni et al., 2021). GPP and LAI
data was retrieved from the Integrated Climate Data Centre (ICDC) website (http://icdc.cen.uni-hamburg.de/las/). The MODIS GPP and LAI data mostly have four observations per month. We took the average of the observations in a month and compared them against the monthly averaged CLM5 data. We compared the MODIS monthly spatial observations with corresponding CLM5 simulations from 2001 to 2014. This exercise is to observe the spatial variation in LAI and GPP over the Indian region. We also compared the spatially averaged time series of monthly LAI and GPP over the Indian subcontinent for the period 2001
to 2014. This exercise is to compare the inter annual cycle in MODIS observations and CLM5 simulations.

### 2.6.4 Latent and Sensible Heat Flux

For the evaluation of changes in surface energy fluxes, we used the FLUXCOM data (Tramontana et al., 2016; Jung et al., 2019). FLUXCOM data is generated using machine learning to merge the flux measurements in eddy covariance towers with remote sensing and meteorological data and estimate surface fluxes (Jung et al., 2019). We used the monthly 0.5° resolution



RS_METEO version of the FLUXCOM data for comparison against the CLM5 simulations. We compared the monthly spatial average of heat fluxes against CLM5 simulations. We also compared the inter-annual time series of heat fluxes with the CLM5 simulations.

## 3 Results

### 3.1 Outcomes of model improvements at site scale

#### 3.1.1 Spring wheat

##### 3.1.1.1 LAI

The Leaf Area Index (LAI) impacts biomass accumulation and transpiration process, while biomass distribution directly affects the yield. Furthermore, LAI is crucial in modeling multiple processes including evapotranspiration and canopy photosynthesis. Additionally, it impacts the dimensions of the contact between the plant and the atmosphere, which is crucial in estimating the

transfer of energy and matter between the canopy and the atmosphere (Su et al., 2022). Therefore, LAI is the most important of the three variables evaluated here.

Figure 3 depicts the time series of LAI simulated by the three different versions of CLM5 for different sites. Results show that CLM5_Def simulated wheat growth during April-June while CLM_Mod1 and CLM_Mod2 simulated wheat growth in November-March. Clearly, the CLM5_Def simulated the wheat growth in the wrong season compared to observations.

Furthermore, CLM5_Def also underestimated LAI. The seasonality error is corrected in CLM5_Mod1 with the change in the sowing window (min_ and max_NH_planting_date in Table 1), but it still underestimated LAI. Including latitudinal variation in base temperature in the CLM5_Mod2 case improved the LAI simulation by reducing the underestimation in most sites except Cooch Behar (Figure 3(a) and 3(b)), Faizabad (Figure 4(c), 4(d) and 4(e)), and a few growing seasons in Naida (Figure 4(o)). Overall, CLM5_Mod2 provided the best estimates of LAI (Fig. 4).

Table 3 shows the impact of improvements made to the CLM5 model. The observed mean maximum LAI is 4.22 m$^2$/m$^2$. CLM5_Mod2 is the closest to the observation with a value of 3.47 m$^2$/m$^2$, while CLM5_Def is the worst with a value of 2.36 m$^2$/m$^2$. Figure 3 shows us that the crop in the CLM5_Def case grows in the wrong season than what is observed. Hence, all performance metrics for the LAI simulations in the CLM5_Def case will show very poor results because the simulated LAI values are all zero during the observed growing season. To ensure a fairer comparison, between the CLM5_Def and

CLM5_Mod cases, we used days from sowing instead of calendar dates in the LAI time series. Even after adjusting for the growing season, the LAI in the CLM5_Def case has a large MAB of 0.81. The CLM5_Mod1 and CLM5_Mod2 performed much better with MABs of 0.52 and 0.43. The negative r-value for LAI in the case of CLM5_Def is due to the simulation of smaller growing lengths and having zero LAI values when the observations reach their maximum values. The r-value improved in both the Mod cases, with a higher *r*-value of 0.3041 (significant at p<.01) in the CLM5_Mod2 case. KGE value is a good

measure of how the model is performing both in seasonality and spatially. KGE for CLM5_Def is very low (-0.62). CLM5_Mod1 showed improvement with a value of -0.02 but it is still negative. CLM5_Mod2 has the highest value of 0.19. The improvement in LAI simulations is also evident from the Taylor diagram in Figure 4(a). The LAI simulations in Mod





cases are closer to the observational value shown in the black star. Overall, the modified models showed significant improvement over the default, with CLM5_Mod2 performing the best.




**Figure 3: Site scale LAI simulated by three versions of CLM5 against observations for spring wheat.**

### 3.1.1.2 Yield

The observed mean yield is 3.88 t/ha (Table 3). The default model underestimated the mean yield with a value of 3.05 t/ha.

The modified models performed better with both producing 3.68 t/ha. All metrics showed that the default model is the worst performer with high MAB and RMSE and low correlation and KGE values. The CLM5_Mod1 is the best performer in all metrics. This is also evident in the Taylor diagram (Fig. 4(b)). However, it is important to note that CLM5_Mod2 performs quite well. Their mean yields are identical and the correlation values of 0.38 in CLM5_Mod1 and 0.30 for CLM5_Mod2 are not different in a statistically significant (p<0.05) way.

**3.1.1.3 Growing Season Length**



The growing season length simulated by CLM5_Def is very low with a mean growing season of just 69 days, compared to 129 days in observations (Table 3). The growing season length considerably increased to 126 days in CLM5_Mod1 and 136 days in CLM5_Mod2. The MAB in the growing season length in CLM5_Mod1 and CLM5_Mod2 are 0.11 and 0.10, respectively, which are much lower than the 0.47 in the CLM5_Def case. Similarly, the modified models performed significantly better than

the default in terms of the other metrics. Their performances are comparable with no statistically significant difference (p<0.05) between the metrics. The Taylors diagram (Figure 4(c)) showed that although CLM5_Mod1 and CLM5_Mod2 have similar correlation, CLM5_Mod2 is performing slightly better with a lower standard deviation and a range comparable to that of the observations. Therefore, the CLM5_Mod2 performed slightly better than CLM5_Mod1 in simulating the growing season length of wheat crop.

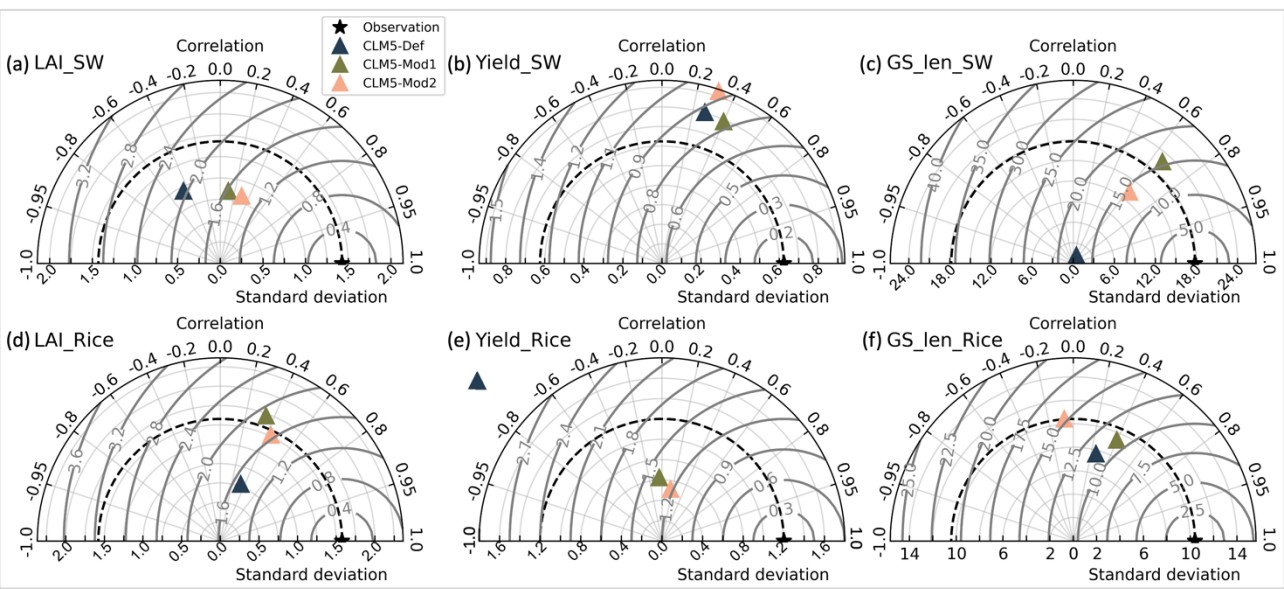


**Figure 4: Taylor diagrams of site-scale CLM performance against observations. (a-c) Spring wheat; (d-f) rice.**

The results in spring wheat showed that both the LAI and growing season length have significant improvement in CLM5_Mod2 over CLM5_Mod1. Table S3 expands on the results discussed above to separately show the improvements observed during the calibration and validation stage. Based on the overall bias in Table 3 and Table S3, we find that the simulation of wheat

improved largely from default to Mod2.

### 3.1.2 Rice

### 3.1.2.1 LAI

The improvement in LAI simulations of rice, especially after introducing the latitudinal variation in base temperature can be seen in Figure 5 and Table 3. CLM5_Def underestimated the mean maximum LAI with a value of 1.65 $m^2/m^2$, which is much

lower than the observed 5.29 $m^2/m^2$. The modified models perform much better, simulating maximum LAI in the range of





4.45-4.5 m²/m². We compared the LAI performance after correcting for the difference in growing season in CLM5_Def as discussed in Section 3.1.1.1. The MAB reduced from 0.66 in the CLM5_Def case to 0.387 in the CLM5_Mod1 case to 0.343 in the CLM5_Mod2 case. CLM5_Mod2 LAI performed better than CLM5_Mod1 in other metrics- RMSE, *r*-value, and KGE (Table 3) and the improvement is significant at p<.05. The Taylor diagram (Figure 4(d)) showed the improved performance in CLM5_Mod2, with the CLM5_Mod2 mark within the standard deviation of observations and higher *r*-value. Overall, the CLM5_Mod2 is the best performer in simulating LAI.

**Table 3: Evaluation of spring wheat and rice across three CLM5 setups at site scale**

| Parameter | Evaluation Metrics | Wheat | | | | Rice | | | |
|---|---|---|---|---|---|---|---|---|---|
| | | Obs | CLM5_Def | CLM5_Mod1 | CLM5_Mod2 | Obs | CLM5_Def | CLM5_Mod1 | CLM5_Mod2 |
| **LAI (m²/m²)** | Mean of max. LAI | 4.22 | 2.36 | 2.69 | **3.47** | 5.29 | 1.65 | **4.48** | 4.45 |
| | MAB | -- | 0.81 | 0.52 | **0.43** | -- | 0.66 | 0.39 | **0.34** |
| | RMSE | -- | 2.61 | 1.76 | **1.41** | -- | 3.00 | 1.94 | **1.68** |
| | r | -- | -0.45* | 0.11 | **0.30*** | -- | 0.34* | 0.34* | **0.43*** |
| | KGE | -- | -0.62 | -0.02 | **0.19** | -- | -0.06 | 0.33 | **0.42** |
| **Yield (t/ha)** | Mean | 3.88 | 3.05 | **3.68** | **3.68** | 4.56 | 2.62 | **3.51** | **3.43** |
| | MAB | -- | 0.25 | **0.15** | 0.19 | -- | 0.70 | **0.30** | **0.29** |
| | RMSE | -- | 1.19 | **0.77** | 0.93 | -- | 3.82 | 1.70 | **1.64** |
| | r | -- | 0.27 | **0.38** | **0.30** | -- | -0.76* | -0.04 | **0.16** |
| | KGE | -- | 0.12 | **0.26** | 0.10 | -- | -1.06 | -0.17 | **-0.04** |
| **Growing season length (days)** | Mean | 129 | 69 | **126** | 136 | 117 | **114** | 123 | 121 |
| | MAB | -- | 0.47 | **0.11** | **0.10** | -- | **0.07** | **0.08** | **0.10** |
| | RMSE | -- | 62.84 | **15.62** | **15.44** | -- | **11.3** | 12.02 | 15.24 |
| | r | -- | 0.37 | **0.66*** | **0.62*** | -- | 0.25 | **0.40** | -0.07 |
| | KGE | -- | -0.21 | **0.57** | 0.52 | -- | 0.21 | **0.39** | -0.07 |
| | Overall bias | -- | 0.51 | 0.26 | **0.24** | -- | 0.48 | 0.26 | **0.25** |

\* significant at p<.05 using the students t-test. The bold font indicates the best performer in each category. If multiple models are marked in bold font, that indicates a lack of statistically significant difference between them.

CLM5_Mod2 simulations performed better in sites in southern (Figures 5(a), 5(b), and 5(c)) and the northern parts of India (Figures 5(g), 5(i), and 5(j)). This strongly suggests that latitudinal variation in base temperature implemented in the CLM5_Mod2 is necessary to capture the growth variation in LAI observed across Indian rice growing regions.

**3.1.2.2 Yield**

The CLM5_Def yield of 2.62 t/ha is much lower than the observed 4.56 t/ha (Table 3). The mean yield improved by nearly 1 t/ha in the CLM5_Mod runs but is still lower than observations. The MAB showed improvement from 0.699 in the CLM5_Def case to 0.297 in the CLM5_Mod1 case and further to 0.291 in the CLM5_Mod2 case. The most significant improvement from CLM5_Def to CLM5_Mod cases is seen in rice yield predictions in the Taylor diagram (Figure 4(e)). RMSE improved from 1.63 t/ha in CLM5_Def to 0.65 t/ha in CLM5_Mod1 and 0.53 t/ha in CLM5_Mod2. Similarly, *r*-value improved from -0.76





in CLM5_Def to -0.04 in CLM5_Mod1 and 0.16 in CLM5_Mod2. KGE has the best value of -0.04 in CLM5_Mod2, which is
far from perfect but is much better than -1.06 in CLM5_Def, and -0.17 in CLM5_Mod1. The improvement from CLM5_Mod1
to CLM5_Mod2 is significant (p<.01) especially in terms of *r*-value and KGE.

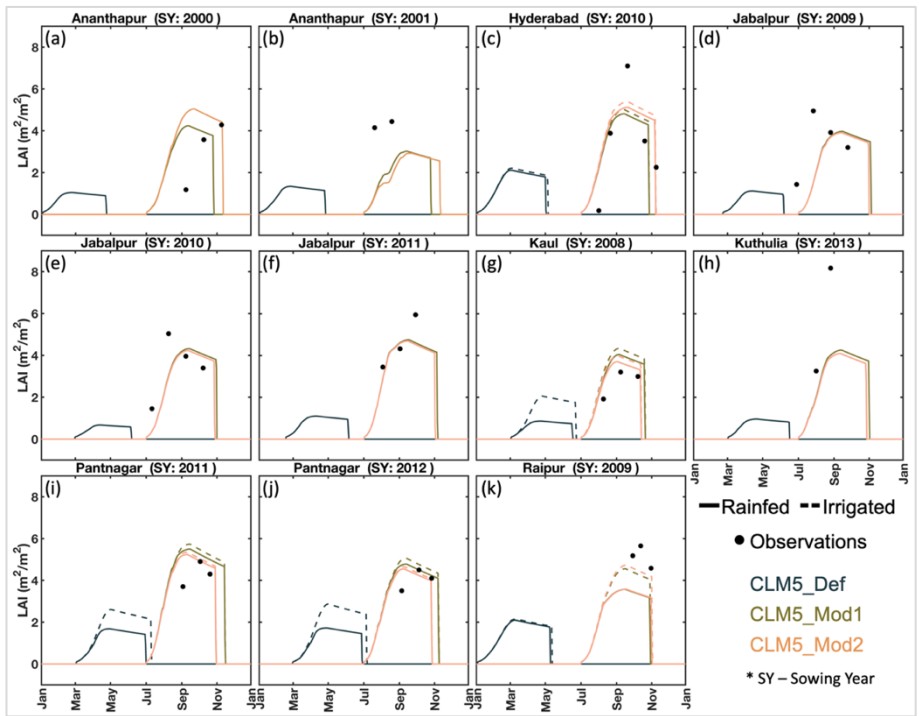

**Figure 5: Site scale LAI simulated by three versions of CLM5 against observations for rice.**

### 3.1.2.3 Growing Season Length

The CLM5_Def model performed quite well in simulating the growing season length with a value of 114 days, which is closest
to the observed value of 117 days (Table 3). The MAB and the RMSE in the default case are the lowest even though the MAB
shows no significant difference among the three CLM5 versions. During our bootstrap exercise with 10000 samples, not a
significant difference between MAB among the three setups is observed. RMSE in CLM5_Mod1 is lower than CLM5_Mod2.
The *r*-value in CLM5_Mod2 (-0.07) shows no variation in growing season length among the sites. However, Figure 5 shows
that the longer or shorter growing season lengths observed at the site scale are simulated in CLM5_Mod2. The Taylor diagram
(Figure 4(f)) showed that the CLM5_Mod1 performs slightly better than the CLM5_Def case and much better than the
CLM5_Mod2 case.

The overall bias in Table 3 for rice shows that the CLM5_Mod2 is performing significantly better than the other CLM5
versions.  This strongly suggests that latitudinal variation in base temperature implemented in the CLM5_Mod2 is necessary
to capture the growth variation observed across Indian rice growing regions.

### 3.2 Outcome of model improvements at the regional scale





### 3.2.1 Yield

Figure 6 compared regional-scale yield simulations by CLM5 against the EarthStat data (Ray et al., 2012). CLM5_Def simulations underestimated the spring wheat yield in central and south-central regions of the wheat growing regions, which is

also identified by Lombardozzi et al. (2020). In the CLM5_Mod1 case, the underestimation found by Lombardozzi et al. (2020) reduced, but at the same time, an overestimation of yield is observed in the eastern parts of the wheat-growing regions. This is rectified to some extent by introducing latitudinal variation in the CLM5_Mod2 case. Large parts of the wheat-growing regions have a low bias between -1 and 1 ton per hectare compared to the EarthStat data. One important region where CLM5_Mod2 is underestimating is the Punjab and Haryana regions (the northwest region in the map). In Figure S8 we compared the total

annual yield from wheat growing regions simulated by CLM5 with FAO data. CLM5_Mod1 replicates the trend observed in FAO data. CLM5_Def clearly underestimated the total yield owing to the lower growing season simulated in the default case.

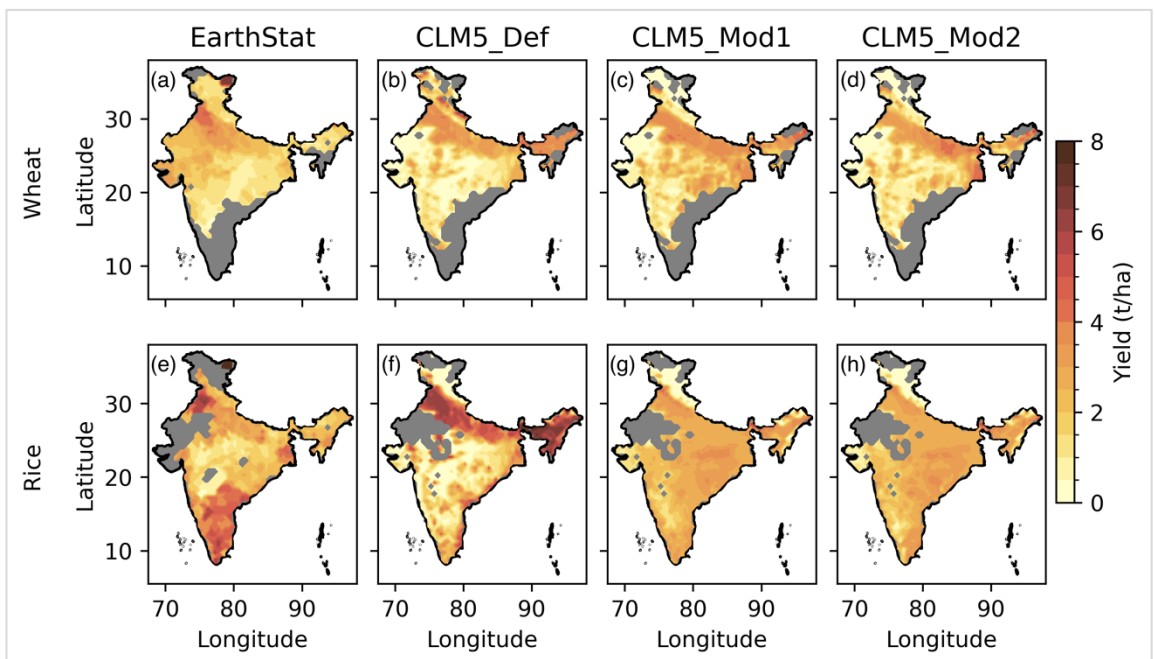

**Figure 6: (a) and (e) yield from EarthStat 2005 for wheat and rice, respectively. (b - d) difference in wheat yield between CLM5 (mean 2003-2007) versions and EarthStat wheat data. (f - h) the difference in rice yield between CLM5 (mean 2003-2007) versions**
**and EarthStat rice data.**

The CLM5_Def rice simulations underestimated the yield across large parts of the rice-growing regions and overestimated it in the Indo-Gangetic Plains (IGP) and northeast regions. CLM5 simulated higher yield in IGP, which has a comparatively lower rice growing area compared to central and eastern parts of India (Figure S7). Improved yield simulation is observed in the CLM5_Mod1 case due to changes in the growing season and grain fill threshold. The overestimation in IGP and the

underestimation in southern parts of India decreased (Figure 6(g)). However, changes made in the CLM5_Mod2 case showed little improvement in most regions over the CLM5_Mod1 case (Figure 6(g) and 6(h)). In CLM5, rice is grown only during the Kharif season; however, in the southern regions of India, where water is available throughout the year, rice is grown in two or



three seasons (Wang et al., 2022). Therefore, the annual yield observations in EarthStat are higher in this region and are not reflected in the CLM5 simulations. In Figure S8(b), we compared the annual rice yield over rice-growing regions of India from

CML5 simulations and FAO data. CLM5_Def overestimated the yield considering the fact that rice is growing in only one season in CLM5. With the improvements made in CLM5, the trend in FAO is matched by the modified simulations, however, yield in modified cases is lower compared to FAO data across the fifteen-year period. The underestimation in yield is expected due to the same reason that rice is growing in only one season in CLM5.

The underestimation of yield for wheat and rice pointed out by Lombardozzi et al. (2020) is reduced to some extent with the

modifications in this study. Bias in yield, especially in rice, is around ±3 t/ha in the default case, which is reduced in CLM5_Mod2 to ±1.5 t/ha. However, more research is required to understand the reason for the bias in CLM5_Mod cases in the range of ±1.5 t/ha in both rice and wheat.

**3.2.2 Irrigation**

We compared our simulated irrigation across wheat and rice-growing regions of India against the annual irrigation patterns

from Biemans et al. (2016). In Figure 7, the blue line shows the annual irrigation pattern simulated by Biemans et al. (2016), the black line depicts irrigation simulated by the CLM-Def case, and the green and orange lines show the CLM5_Mod1 and CLM5_Mod2 simulations, respectively. CLM5_Def has anomalous peaks in the pre-monsoon summer season for both wheat and rice. These are also found in Mathur and AchuthaRao. (2019). This error in irrigation seasonality resulted from wrong cropping patterns of wheat and rice in India in the CLM5_Def case. The modified CLM5 simulations matched the patterns

from Biemans et al. (2016). One major difference between the current study and Biemans et al. (2016) is that the rice is grown in the rabi and kharif seasons in Biemans et al. (2016), while in our study, rice is sown in only the Kharif season. CLM5 is not currently equipped to simulate multiple crop sowings in a year, and the rainfed and irrigated rice crop maps of CLM5 (Figure S7) do not reflect the kharif and rabi rice crop maps. Another important point to note is that Biemans et al. (2016) reported the total irrigation water demand of the crop during the growing season, and we are comparing it with water added through

irrigation to the crops.

The improvements made in our study improved the seasonality of the irrigation in wheat and rice croplands. The improved models simulate less water added through irrigation for the wheat and rice crops. Water added through irrigation over the wheat growing region is reduced from 4.32 billion cubic meters/day (BCM/day) in CLM5_Def to 3.08 BCM/day in CLM5_Mod1 and 3.53 BCM/day in CLM5_Mod2. The drastic difference in irrigation water added is because wheat is now

growing in the rabi season in the Mod cases compared to the summer season in CLM5_Def. A greater reduction in irrigation water added to crops is observed in the case of rice. CLM5_Def simulates 8.09 BCM/day of water added through irrigation, while CLM5_Mod1 and CLM5_Mod2 simulates only 2.97 and 3.09 BCM/day, respectively. Such drastic differences in water added through irrigation will greatly impact the hydrological cycle.





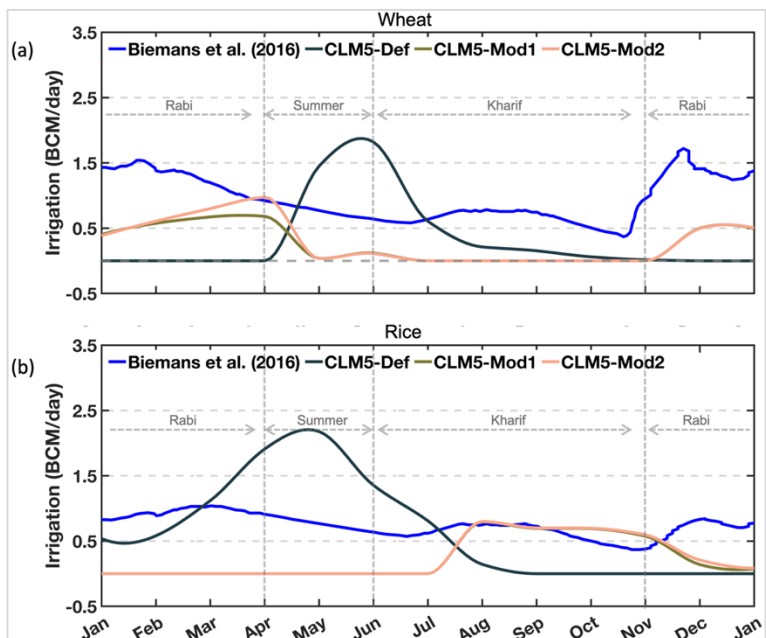

**Figure 7: Comparison of water added through irrigation simulated by CLM5 and water demand data from Biemans et al. (2016).**

### 3.2.3 GPP and LAI

#### 3.2.3.1 Spatiotemporal variation

The monthly spatial patterns of simulated GPP and LAI are showed in Figures 8 and 9. The major crop-growing months are June till March. This is evident in the MODIS GPP and LAI observations. However, the CLM5_Def simulated low GPP and LAI during this period. This is due to the error in the crop calendar in the default model. CLM5_Def simulated maximum carbon uptake (GPP) and LAI in April and May (Figure 8: Apr and May) when very little vegetation activity is observed across India, which is also evident from MODIS GPP and LAI data (Figure 9: Apr and May). In contrast, the modified models simulated the GPP and LAI cycle as observed in the MODIS data with high GPP and LAI during June-March and low values during the rest of the year.

The maximum observed GPP in the MODIS data is in the northeast and peninsular regions of India. In contrast, the maximum GPP simulated by CLM5_Def is in the IGP region. The CLM5_Mod1 and CLM5_Mod2 simulations are similar to the MODIS observations with maximum LAI in central and eastern parts of the country from July to February months of the year. Even though the modified models captured the observed spatial patterns, they tend to overestimate the magnitudes.




**Figure 8: Spatial variation of GPP simulated by CLM5 against MODIS data. The data shows the monthly GPP averaged over 2000-2014.**

### 3.2.3.2 Monthly time series

We evaluated the monthly time series of GPP and LAI from 2000 to 2014 (Table 4; Figures S2). The simulated GPP performed better in the modified versions of CLM5 than the default one. The monthly mean GPP has an MAB of 0.51 in CLM5_Def, 0.241 in CLM5_Mod1, and 0.235 in CLM5_Mod2. The RMSE decreased from 6.95 kgC/m$^2$/mon in CLM5_Def to 3.48 kgC/m$^2$/mon in Mod1 and 3.56 kgC/m$^2$/mon in Mod2. The most significant improvement in the model simulations is seen in the correlation of CLM5 simulated GPP against the MODIS observations. The *r*-value is negative in the case of CLM5_Def (-

0.47) because the seasonality of vegetation growth in the Indian region is incorrect. The *r*-value improved to 0.76 in CLM5_Mod1 and CLM5_Mod2. Similarly, KGE has a negative value (-0.48) in CLM5-Def and improved to 0.72 in CLM5_Mod1 and 0.71 in CLM5_Mod2.







**Figure 9: Spatial variation of LAI simulated by CLM5 against MODIS data. The data shows the monthly LAI averaged over 2000-**
**2014.**

The peaks in annual GPP from 2001 to 2014 (in Figure S3(a)) in the case of CLM5_Def are off by at least three months compared to MODIS GPP, while the peaks in CLM5_Mod1 and CLM5_Mod2 are consistent with the observations. Figure 10(b) shows the monthly GPP comparison of CLM5 simulations against MODIS data in a Taylor Diagram. A drastic improvement is observed from default to modified cases; the correlation improved along with standard deviation, which got
very close to observations (black star on Taylor Diagram) in the modified cases. CLM5_Mod2 is the best-performing setup in Figure 10(b), with high correlation and low standard deviation.

Interestingly, not all evaluation metrics for LAI improved with changes made to CLM5 in this study. The monthly mean LAI had an MAB of 0.19 in the CLM5_Def case, 0.24 in the CLM5_Mod1 case, and 0.3 in the CLM5_Mod2 case. RMSE in CLM5_Def is 0.27 $m^2/m^2$, which increased to 0.29 $m^2/m^2$ in the CLM5_Mod1 case and 0.35 $m^2/m^2$ in the CLM5_Mod2 case.
The overestimation of LAI is consistent across all CLM5 simulations (Figure S2(b)). The overestimation of LAI by process-





based vegetation models compared to MODIS LAI data is widely reported (Fang et al., 2019). The reasons are processes like carbon fixation and allocation of biomass to leaves in the models (Gibelin et al., 2006; Richardson et al., 2012), differences in defining the LAI by various models and MODIS (Fang et al., 2019), and due to inherent bias in LAI estimation in MODIS in the equatorial region (20 °S to 15 °N) (Fang et al., 2019; Lin et al., 2023). Figure S3(b) illustrates that although the bias is

higher in Mod cases, the peaks in annual LAI in MODIS data are captured accurately by the Mod cases. The CLM5_Def peak in LAI is off by two to three months.

**Table 4: Evaluation of CLM5 simulations at the regional scale against MODIS (LAI and GPP) and FLUXCOM (LH and SH) data. The bold text states that the version of CLM5 is performing the best.**

| Parameter | Evaluation Metrics | CLM5_Def | CLM5_Mod1 | CLM5_Mod2 |
|---|---|---|---|---|
| GPP | MAB | 0.51 | 0.24 | **0.24** |
| | RMSE | 6.95 | **3.48** | 3.56 |
| | $r$ | -0.47* | 0.76* | **0.76*** |
| | KGE | -0.48 | **0.72** | 0.71 |
| LAI | MAB | **0.19** | 0.24 | 0.31 |
| | RMSE | **0.27** | 0.29 | 0.35 |
| | $r$ | 0.35* | 0.92* | **0.93*** |
| | KGE | 0.34 | 0.40 | **0.41** |
| LH | MAB | 0.22 | 0.17 | **0.16** |
| | RMSE | 14.78 | 11.91 | **11.28** |
| | $r$ | 0.69* | 0.93* | **0.93*** |
| | KGE | 0.60 | 0.77 | **0.77** |
| SH | MAB | 0.22 | **0.19** | 0.20 |
| | RMSE | 14.34 | **11.16** | 11.56 |
| | $r$ | 0.85* | 0.94* | **0.95*** |
| | KGE | 0.52 | 0.73 | **0.73** |

* significant at p<.01 using the students t-test

Other evaluation metrics of LAI showed that the modified models are performing much better than the default case. The $r$-value in CLM5_Def is 0.35, which increased to 0.92 in the CLM5_Mod1 case and 0.93 in the CLM5_Mod2 case. Higher $r$ values in modified runs imply that the seasonality of LAI simulated by CLM5 considerably improved due to the improvements made in the model. KGE metric showed improvement from 0.35 in the CLM5_Def case to 0.4 in the CLM5_Mod1 case and to 0.41 in the CLM5_Mod2 case (Table 4). The Taylor diagram of LAI (Figure 10(a)) showed improvement in correlation, but

the error and standard deviation are higher than the observations.

### 3.3 Heat fluxes

### 3.3.1 Latent Heat flux

### 3.3.1.1 Spatial variation

The spatial and monthly variation in the CLM5 simulation of LH is illustrated in Figure S4. Most of the spatial pattern in

observed LH is captured by all setups of the CLM5 model. One error, though, in the case of CLM5_Def, is observed in March, April, and May, where the IGP region shows high LH values that are absent in FLUXCOM observations. The reason for this





erroneous high LH in this region is due to the growth of wheat evident from Figure 9. The least LH is observed during the winter months, November to February, across all CLM5 simulations.

### 3.3.1.2 Monthly time series

Comparing the latent heat flux (LH) simulated by CLM5 with FLUXCOM data, we observe MAB of the LH reduced from 0.22 in CLM5_Def to 0.27 in CLM5_Mod1 and 0.16 in CLM5_Mod2. The RMSE reduced from 14.74 W/m$^2$ in the CLM5_Def to 11.91 W/m$^2$ in CLM5_Mod1 and 11.28 W/m$^2$ in CLM5_Mod2. The correlation improved from 0.69 in CLM5_Def to 0.93 in CLM5_Mod1 and CLM5_Mod2 cases. KGE metric improved from 0.70 in CLM5_Def to 0.77 in CLM5_Mod cases. The improvement is evident in the Taylor diagram (Figure 10(c)), where CLM5_Mod simulations are much closer to the

observations than the CLM5_Def case. CLM5_Mod1 and CLM5_Mod2 have similar performance, even though LAI improved in CLM5_Mod2 over CLM5_Mod1. Figure S5(a) depicts that the CLM5 simulations underestimate the LH compared to FLUXCOM data.

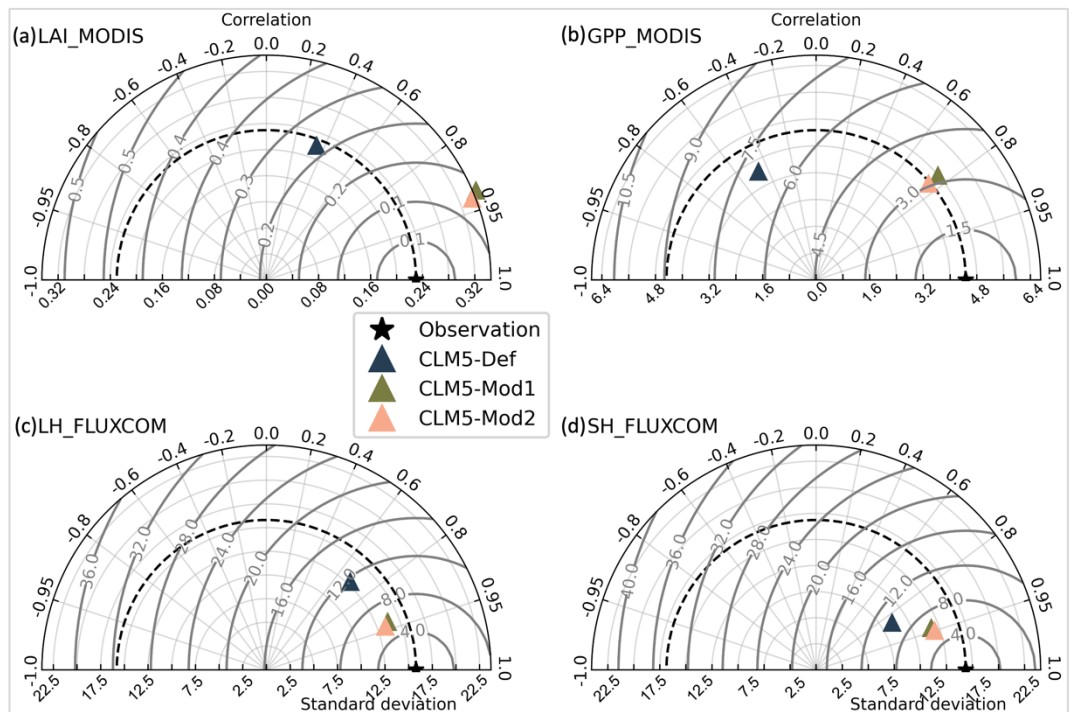

**Figure 10: Comparing CLM5 simulated (a) LAI, (b) GPP, (c) LH, and (d) SH against observations. The data used here is the monthly**
**mean from 2000 to 2014.**

### 3.3.2 Sensible Heat flux

### 3.3.1.1 Spatial variation

The spatial and monthly variation in the CLM5 simulation of SH is illustrated in Figure S5. Most of the spatial pattern in observed SH is captured by all setups of the CLM5 model. However, CLM5_Def simulated slightly lower SH compared to



the modified model simulations, especially during the months of March to June. Low SH is observed during the months of August to December across all CLM5 simulations.

### 3.3.2.2 Monthly time series

Comparing the sensible heat flux (SH) simulated by CLM5, we observed the MAB of SH reduced from 0.22 in CLM5_Def to 0.19 in CLM5_Mod1 and 0.20 in CLM5_Mod2. The RMSE reduced from 14.34 W/m$^2$ in CLM5_Def to 11.16 W/m$^2$ in

CLM5_Mod1. The RMSE in CLM5_Mod2 is 11.56 W/m$^2$, slightly higher than in the CLM5_Mod1 case. The correlation improved from 0.85 in CLM5_Def to 0.94 in CLM5_Mod1 and 0.95 in CLM5_Mod2. KGE metric improved from 0.52 in CLM5_Def to 0.73 in CLM5_Mod cases. The SH in CLM5 is affected by vegetation temperature and ground temperatures. The results suggest that a difference in vegetation temperatures is observed between CLM5_Def and CLM5_Mod1, very little to no difference is observed from CLM5_Mod1 to CLM5_Mod2. The difference in vegetation temperature is likely caused by

the accurate representation of the growing season in CLM5_Mod cases compared to CLM5_Def. This is also evident from the Taylor diagram (Figure 10(d)), where we see improvement from CLM5_Def to CLM5_Mod1, but CLM5_Mod1 and CLM5_Mod2 markers overlap. Figure S5(b) depicts that the CLM5 simulations underestimated the highs and lows of SH in FLUXCOM data. The peak of SH in all CLM5 simulations is in line with the FLUXCOM data. However, CLM5_Def has a larger bias in estimating the maximum SH during a year.

Overall, the improvements in the representation of the two major Indian crops drastically improved the surface energy flux simulations by CLM5.

### 4 Discussions

In this study, we improved the representation of spring wheat and rice, the two major crops grown in India, in the CLM5 land model. One major strength of the current study is using multiple site-scale observations for calibrating and validating the crop

modules in CLM5. Studies such as those by Gahlot et al. (2020), who looked at Indian crops, used only one site for calibrating and evaluating their model. Even studies carried out for winter wheat across the globe (Lokupitiya et al., 2009; Lu et al., 2017; Boas et al., 2021) used two or three sites for calibrating the model. In contrast, we used 33 growing seasons from 14 sites resulting in a very rigorous calibration and evaluation exercise. The improved model in our study not only simulated crop phenology better but also improved the simulation of energy and water fluxes. The results demonstrate the importance of

accurate representation of crops in land surface models, especially in a country like India, where more than 50% of land is used for agriculture.

This study looked at the variability in yield simulations at a regional scale for two major Indian crops. When compared against the EarthStat 2005 yield data, few regions showed improvement from the default CLM5 version to the modified version. Nevertheless, the yield simulated by CLM5 for wheat and rice needs improvement. Yield is now calculated as the available

dry matter allocated to the grain after the allocation to root, leaf, and stem. Global studies like Rabin et al. (2023) highlighted the issue of inconsistent improvement in yield estimates at different scales while analyzing the inter-annual and spatial variation in yield estimates. A recent study by Yin et al. (2024), which looked at the yield estimates by various models concluded that



the CLM5 simulated the temporal variability well but failed to simulate the spatial variability across China's wheat and rice growing regions. Similarly, in our study, we found an improvement in site-scale yield estimates over different growing seasons but found mixed results in regional yield estimates. The yield should perform better since the CLM5 simulates the GPP with lower bias and improved seasonality. However, that is not the case here. Therefore, an investigation into the yield estimation, especially wheat in CLM5, is necessary.

A region with significant agricultural coverage and practices is misrepresented in the most widely used land surface model. Our study improved the model representation of the two major Indian crops. Our future goal is to study the feedback in the land-atmosphere system using the improved land model. The enhanced crop representation and its management practices will impact the water cycle and local and global temperature and precipitation (Mathur and Rao, 2020). Rice and wheat constitute 80% of India's harvested land area, followed by maize, sugarcane, and cotton. Improving parameterizations for all these Indian crops (seasonal and cash crops) would be an ideal next step.

While our study made progress in correcting shortcomings, it is critical to recognize that the CLM5 model, like any sophisticated climate model, is still a work in progress. Future improvements should address broader model deficiencies highlighted in ours and various other studies. The deficiencies range from the inclusion of sophisticated plant and soil hydraulics (Boas et al., 2021; Raczka et al., 2021), improvement in yield predictions, improved or new management practices like tillage (Graham et al., 2021), post-harvest crop residual management. Furthermore, our research contributes to continuing attempts to improve the CLM5 model by addressing shortcomings in Indian crop representation. The enhancements are a step forward, emphasizing the iterative nature of model development and the importance of constant refinement to ensure the model's accuracy in replicating complex earth system processes. Future studies should build on these findings, including additional enhancements to address broader shortcomings in the model.

## 5 Conclusion

Two major modifications were made to CLM5 in this study. First, the representation of the growing seasons of wheat and rice in India were improved so that they align better with the observations. Second, a latitudinal variation in base temperature was implemented to capture the crop varieties grown across diverse Indian agro-climatic conditions. These modifications resulted in the following improvements in the CLM5 simulations:

- The crop phenology is realistic in the modified models. The models are simulating rice and wheat growth in the seasons they are actually grown in the field.
- The LAI simulations are significantly better in both wheat and rice at the site scale. The bias in the simulations reduced by nearly 50% compared to the default model.
- The simulated growing season length for wheat is significantly better at the site scale. The rmse improved from over 60 days in the default model to just over 15 days.
- The simulations of rice yield are significantly better at both site and regional scales.



- The carbon uptake (GPP) simulations over the Indian region are significantly better, improving from a negative correlation in the default model to a high positive correlation.
- The seasonality of simulated irrigation patterns across crop regimes in India is realistic.

Irrigation is a major part of agriculture in India. With the improvements made to the model, irrigation patterns improved drastically and are now in line with a study by Biemans et al. (2016). The amount of water taken up by the crops through irrigation during their respective growing seasons decreased, and at the same time, the latent heat simulations improved from the default case.

CLM5 defines its crop parameters globally and, therefore, ends up having a large bias in regions such as India, where crop practices are not like those followed in Europe or North America. This study demonstrated that the global land models must use region-specific parameters rather than global parameters for accurately simulating vegetation and, therefore, land surface processes. Such improved land models will be a great asset in investigating the global and regional scale land-atmosphere interactions and developing improved future climate scenarios.

**Code and data availability:** The site scale data used in the study is available at Varma et al. (2024). The code changes made in CLM5, domain, surface, and land use time series data used for the site scale and regional simulations are available at Reddy et al. (2024).

**Author Contribution:** KNR and SBR conceptualized the study. KNR, SBR, SSR, and DLL designed the methodology. KNR conducted the experiments. KNR made changes to the code with the guidance from SSR. GVV and RB compiled the site scale data. RB conducted a few site-scale experiments. DeCN generated CLM input files required for experiments. KNR analyzed the results and wrote the manuscript. SSR reviewed the manuscript. SBR edited the paper.

**Competing Interests:** At least one of the (co-)authors is a member of the editorial board of Geoscientific Model Development.

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
