# Peer review of "Improving the representation of major Indian crops in the Community Land Model version 5.0 (CLM5) using site-scale crop data"

_EGUsphere, 2024_

## Referee Comment (RC3)

**Review of** "Improving the representation of major Indian crops in the Community Land Model version 5.0 (CLM5) using site-scale crop data, K. Narender Reddy, Somnath Baidya Roy, Sam S. Rabin, Danica L. Lombardozzi, Gudimetla Venkateswara Varma, Ruchira Biswas, and Devavat Chiru Naik"

**Summary**

The article focuses on improving the representation of two major Indian crops, spring wheat and rice, in the Community Land Model version 5.0 (CLM5) to enhance its accuracy in simulating crop phenology, yield, and associated land-atmosphere interactions. Using a newly created, comprehensive site-scale crop data set from India, the study calibrated and adjusted key parameters in the CLM5 crop module, such as sowing dates, growth parameters, and base temperature. The modified model versions (CLM5_Mod1 and CLM5_Mod2) demonstrated significant improvements in simulating crop growth, water and carbon fluxes, and irrigation patterns compared to the default CLM5 version. These modifications underscore the importance of region-specific parameters for global land models and provide a basis for better understanding land surface processes and their role in climate scenarios. The study's findings have implications for regional agricultural management and policy, as well as for enhancing climate modeling accuracy.

**Title**

The title generally works well with the content of the manuscript but mentioning the specific crops worked on in the title would provide readers with some clarity on what to expect.

**Abstract**

The abstract provides a concise summary of the study's goals, methodology, and findings, stating that the modified CLM5 performs better in simulating crop phenology, yield, and fluxes. For instance, it mentions that the Pearson's $r$ for monthly LAI improved from 0.35 to 0.92 and monthly GPP from -0.46 to 0.79 compared to MODIS data.

While the abstract states that it aims to improve the representation of Indian crops, it could be more specific earlier on by naming the two crops (spring wheat and rice). This would immediately inform the reader about the study's focus. Consider revising the sentence: "Our study aimed to improve the representation of these crops in CLM5" to "Our study aimed to improve the representation of spring wheat and rice in CLM5."

The abstract could briefly mention the broader implications of these improvements. For example, it could say, "These improvements can enhance the accuracy of land-atmosphere interaction studies and inform regional agricultural management and policy."

**Introduction**

The introduction effectively outlines the importance of accurately simulating cropland processes in Land Surface Models (LSMs), which impact energy, water, and carbon fluxes. It provides sufficient background on the Community Land Model (CLM) and its development up to version 5.0.

While the introduction cites several relevant studies (e.g., Elliott et al., 2015; Lombardozzi et al., 2020), it could benefit from a few more recent references to highlight the current state of crop modeling. For example, "Recent studies provide valuable insights for enhancing the accuracy of simulating biogeophysical and biogeochemical processes..." could include more studies published after 2020 to strengthen this point.

The introduction mentions that "CLM5 simulations of rice and wheat over the Indian subcontinent show large biases," but it could be clearer about what specific biases (e.g., underestimation of yield, incorrect phenology timing) the current study addresses. Adding a sentence such as, "Specifically, the model has been shown to inaccurately simulate the timing of planting and harvesting for spring wheat and rice, leading to incorrect estimates of carbon fluxes and water use," would provide a clearer problem statement.

**Materials and Methods**

The description of the CLM5 model and its modifications (CLM5_Mod1 and CLM5_Mod2) is comprehensive, outlining the data sources, simulation setups, and parameter modifications. The distinction between site-scale and regional-scale simulations is also clearly made.

The detailed methodological description would benefit from a flowchart or diagram summarizing the process, from data collection to model calibration and evaluation. For example, Figure 1 in the document effectively shows the sites used for calibration, but a flowchart could visually represent the steps outlined in Sections 2.1 to 2.3.1.

The manuscript describes various parameter changes (e.g., base temperature, planting dates), but it could provide more justification for selecting these specific parameters for sensitivity analysis. For example, the section "Improvements in CLM5" states, "The base temperature and maximum GDD control the longevity of each phase in crop growth," but it does not explain why these were chosen over other potential parameters. A brief explanation could be added, such as "These parameters were chosen based on their significant influence on phenological development stages in crops, as indicated by previous studies (cite studies)."

**Results**

The results are presented with clear visualizations, such as the Taylor diagrams and time-series plots, that compare model versions against observational data.

While the Taylor diagrams (e.g., Figure 4) effectively show improvements in the model's performance, they could benefit from a brief explanation of how to interpret them. For example, "Higher correlation, lower RMS error, and smaller standard deviation characterize the most accurate CLM5 configuration, as seen in the closer proximity of CLM5_Mod2 markers to the observational reference point."

The results section presents the remaining biases in yield and growing season length (e.g., "The growing season length simulated by CLM5_Def is very low with a mean growing season of just 69 days, compared to 129 days in observations"), but it could discuss potential reasons for these biases in more detail. For example, it could mention model assumptions, data limitations, or unaccounted-for environmental factors that could contribute to these discrepancies.

**Discussion**

The discussion appropriately links the results to the study objectives, emphasizing the importance of region-specific parameters in LSMs for improving crop simulation accuracy.

The manuscript mentions, "Such improved land models will be a great asset in investigating global and regional-scale land-atmosphere interactions and developing future climate scenarios," but it could expand on specific applications. For instance, how could this model be used to inform irrigation management practices or forecast agricultural productivity under different climate scenarios?

The discussion would benefit from a dedicated section on limitations and future directions. For example, the text could state, "While the modified models showed significant improvements, there are still biases that could be addressed by incorporating more diverse site data or accounting for multi-cropping practices in the model," and suggest specific future research that could address these limitations.

**Conclusion**

The conclusion reiterates the key findings and reinforces the need for region-specific model calibration.

The conclusion could be strengthened by including a call for further studies or actions, such as "Future work could focus on extending this modeling approach to other major crops in India or integrating socio-economic factors to better inform policy-making."

**Figures and Tables**

Figures and tables are generally well-presented and labeled, effectively supporting the text.

Consider adding more comparative visuals that summarize the improvements across different metrics and model versions. For instance, a bar chart comparing MAB, RMSE, and Pearson's $r$ values for CLM5_Def, CLM5_Mod1, and CLM5_Mod2 could provide a quick visual reference for readers.

**References**

The references are relevant and extensive, covering a range of studies on LSMs and crop modeling.

Include more recent references (post-2020) to ensure the manuscript reflects the latest advancements in the field. For example, search for recent studies on crop modeling in LSMs that may have incorporated new methodologies or datasets.

**General Comments**

The manuscript is generally well-written with a clear technical style suitable for the target audience. However, some sections could benefit from simplified language to increase accessibility for readers from diverse scientific backgrounds.

The manuscript follows a logical flow, but the Methods and Discussion sections could be further refined for clarity and depth.

**Final Recommendations**

Provide more justification for the choice of specific parameters in the sensitivity analysis and model modifications.

Include a discussion on the potential policy and practical implications of the findings and suggest specific areas for future research.

Consider adding more comparative figures and diagrams to succinctly showcase the differences between model versions and their performance improvements.

By addressing these critiques, the manuscript can be strengthened to ensure a robust and impactful contribution to the field. If you need further assistance with more detailed critiques on specific sections or figures, let me know!

---

## Author Comment (AC1)

**Improving the representation of major Indian crops in the Community Land Model version 5.0 (CLM5) using site-scale crop data**

K. Narender Reddy, Somnath Baidya Roy, Sam S. Rabin, Danica L. Lombardozzi, Gudimetla Venkateswara Varma, Ruchira Biswas, and Devavat Chiru Naik

We thank the referees for their thorough reviews and the editor for allowing us to respond to the referees' comments.

Please add the following collaboration for the co-author Danica L. Lombardozzi- "Department of Ecosystem Science & Sustainability, Colorado State University, Fort Collins, CO, USA". The new authors list and the collaboration must look like:

*"K. Narender Reddy[1], Somnath Baidya Roy[1], Sam S. Rabin[2], Danica L. Lombardozzi[2,3], Gudimetla Venkateswara Varma[1], Ruchira Biswas[1], and Devavat Chiru Naik[4]*

*[1]Centre for Atmospheric Sciences, Indian Institute of Technology Delhi, New Delhi, 110016, India*
*[2]Climate and Global Dynamics Laboratory, National Centre for Atmospheric Research, Boulder, 80307, CO, USA*
*[3]Department of Ecosystem Science & Sustainability, Colorado State University, Fort Collins, CO, USA*
*[4]Department of Civil Engineering, Indian Institute of Technology Delhi, New Delhi, 110016, India"*

Below, we provide a point-by-point response to the referees' comments. The comments provided by the referee are in red color, our responses are in black color, and the proposed changes to the text are in green color.

**Referee Comments I:**

We greatly appreciate your insightful evaluation of the abstract and valuable suggestions for improving its quality. Your feedback is integral to the academic process, and we are grateful for your contribution. The following changes will be made to the existing abstract:

1) **Introduction and Purpose:** The abstract establishes the significance of accurate cropland representation in terrestrial simulations in CLM, effectively focusing on spring wheat and rice, which dominate agricultural land use in India. The introduction is clear and compelling, providing a strong rationale for the study. The impetus for the study is critical and timely as accurate representation of crop functional types in non-temperate regions of the world is an essential and current research concern in many Land Surface, Earth System, and Dynamic Global Vegetation Models (LSMs, ESMs & DGVMs) - largely due to data paucity issues. Improving the accuracy of the representation of cropland in a reputable DGVM like CLM will, therefore, contribute to the field of cropland and plant functional type representation in DGVMs overall. As a side note, it may be useful to mention a key "error" with the previous state of crop modeling in CLM that the study now addresses.

- We thank the referee for the appreciative comments.

2) **Methodology:** The methodology is compelling, as it outlines the creation and utilization of a novel, comprehensive spring wheat and rice database to improve the parameterization of crop phenology, growing season, and simulated yield. The use of eight sites encompassing 20 growing seasons for each crop hints at the robustness of the study. The abstract elucidates how this data was used to calibrate the relevant CLM5 crop functional

types, situating the improvements achieved and mentioned in the results in the appropriate methodological context.

3) **Results:** Results outline specific enhancements to CLM5's performance metrics for simulated crop functional types, with comparisons to alternative datasets (MODIS). Significant improvements in Pearson's r values for various simulated crop features like LAI, GPP, and corresponding energy fluxes provide good evidence that the study objectives were effectively achieved, demonstrating the improved accuracy of the model.

4) **Conclusion:** The impact of the study is clear, accentuating the need for region-specific crop functional type parameterization in global LSMs, ESMs, and DGVMs. Broader implications for modeling land-atmosphere interactions across various climate scenarios add value to the research.

5) **Overall Assessment:** This is a comprehensive abstract that effectively outlines the purpose, methodology, results, and conclusion of the study. It maintains an appropriate balance between brevity and detail, making the abstract both informative and accessible.

- We thank the referee for their appreciation of our work.

6) **Suggestions for Improvement:** The abstract could, however, further clarify the novelty of the dataset that was used. Additional detail on aspects of the data that make it unique or unprecedented would be valuable, in the context of the relevance of the data to the study. Secondly, the abstract could benefit from an additional sentence (or two) that emphasizes the broader implications of the study, addressing, for example, how the improvements made to CLM can be used in practical contexts like climate impact modeling on agricultural land. This will add to the utilitarian relevance of the study. Thirdly, a brief mention of any challenges or limitations (e.g., calibration process or data digitization issues) of the improved model would provide a more rounded perspective of the outcomes of the study.

   a. The representation or modeling of tropical ecosystems has been a challenge due to the low availability of data. This has led to a lack of proper understanding of the tropical ecosystems in terms of the carbon capture abilities and impact of extreme events. Further, the lack of understanding has led to the misrepresentation of tropical ecosystems (e.g., Indian agroecosystems) in major land models. The dataset we have put together in this study is aimed at bridging the gap in understanding tropical ecosystems. The major goal of this study is to encourage more research into identifying and collecting data from unconventional sources for understanding and representing land processes in tropical ecosystems more accurately. The novelty will be highlighted in the abstract by adding the following text to the abstract: "This dataset is the first of its kind, covering 50 years and over 20 sites of crop growth data across tropical region, where data has traditionally been sparse both spatially and temporally."

   b. The following text will be added to the abstract to expand on the broader implications of the study: "The improved CLM5 model can be used to investigate the changes in growing season lengths, water use efficiency, and climate impacting crop growth of Indian crops in the future scenarios. The model can also help in providing estimates of crop productivity and net carbon capture abilities of agroecosystems in future climate."

   c. In India, wheat is grown majorly in one season, but rice is grown multiple times yearly. In southern India, where water is available throughout the year, rice is even grown three

times a year. Our study did not include the multiple sowing of rice in India. This major study limitation is explained in detail in section 2.3.1.2 in the manuscript.

d. Another major positive from the current study is that we made a new dataset available for a highly diverse agroclimatic region. This will motivate other modeling groups (e.g., LPJ-Guess, ISAM, E3SM) to improve the representation of tropical vegetation.

**Referee Comments II:**

General Comments:

Marked improvements for spring wheat and rice in India using the CLM5 land model, largely achieved through the calibration of key crop growth and planting parameters. The growing seasons now better align with observations, addressing the previous errors in the crop calendar. Useful and important work. The introduction of latitudinal variation in base temperature is a valuable and important addition.

- We appreciate the thorough evaluation of the manuscript. Thank you for recognizing the value of the study and its importance in driving the Indian agroecosystems modeling studies.
- Thank you for appreciating the incorporation of latitudinal variation in this study. We will highlight this in the abstract as well.

The replies to the specific comments are as follows:

1) L124: Would adding a day length control on planting date/crop emergence help here?

   a. Day length is an important parameter. It is already used in an implicit way. The planting date is determined by the 10-day running mean temperature after the model verifies if the days are warm enough to plant the crops using the variable $GDD_8$ (Lawrence et al., 2018). Therefore, day length has an impact on planting through 10-day running mean temperature. That is why adding day length as an independent parameter is unlikely to have a strong impact on planting day and crop emergence.

2) Why did the 0.4-grain fill threshold for rice perform poorly, while a 0.65 threshold showed improved results? By making this change, you are effectively decreasing yields and growth. How do you justify that a 0.4 threshold worked well at other sites (original value), but a 0.65 threshold yields better results at the studied sites? This seems to connect with the paragraph on Line 439; it would be interesting to expand on this further.

   a. The reason for using a grain-fill parameter of 0.4 for rice in CLM5 might be due to the data used to calibrate the rice crop. The calibration during the CLM5 model development might have referred to the studies from China, which generally have lower temperatures than India and a longer growing season of more than 160 days (Li et al., 2024). Therefore, a 0.4 grain-fill threshold should be sufficient for the crop to reach the reproductive phenological stage. Since the average temperatures are higher in India, a 0.4 grain-fill threshold is causing the rice crop yields to be higher in the northern parts and lower in the southern parts (Figure 1(b.2) in this document; Figure 6(b.2) of the manuscript). LAI is also very low compared to observations, as seen in Figure 5 of the manuscript.

   b. Additionally, in the default CLM5 model, the rice crop is sown in January-February. This period in India has a drastic contrast in temperature from North to South. Hence, a very low yield was simulated in the southern sites (Figure 1(b.2) in this document:

Anantapur, Hyderabad, Jabalpur, and Kuthulia) compared to the northern sites (Figure 1(b.2) in this document: Faizabad, Kaul, and Pantnagar).

    c. The improvement in rice crop growth and yield is twofold in our study. One is changing the growing season, and the other is the grainfill parameter. We will expand on the need to vary the grain fill parameter of rice in the manuscript. The following text will be added at line 462 of the revised manuscript: "The improvement in rice crop growth and yield is twofold in this study. One is changing the growing season, and the other is the grainfill parameter. A study by Rabin et al. (2023) used the CLM5 model to simulate crop yields of major crops across the globe, and the important point to note here is they used a prescribed calendar, therefore the growing season is accurate for crops in all regions, but did not change the grain fill parameter and used the default value of 0.4. The results for rice yield were very poor compared to the FAO data (Rabin et al., 2023). Therefore, merely changing the growing season would not improve the rice crop yields. Our sensitivity studies with the grain fill parameter showed that the value 0.65 produced better crop growth and yields after changing the growing season."

3) Figure 3, simulated LAI during early season growth is generally much higher than observed. Why is this? Also, are there different Spring Wheat cultivars between sites? This could influence results, so it would be interesting to include this if relevant.

    a. The reason for the higher growth in the early season than in the observations is due to the variation in sowing dates in the observation data. We are defining a single window for sowing the crop, and the date of sowing is solely dependent on the 10-day running mean temperature (Lawrence et al., 2018), but in practice, farmers depend on various triggers to sow their crops. Few regions wait for the harvest of the preceding crop (kharif crop), while others wait for the right temperature or water availability. Due to this discrepancy in sowing dates of observations and CLM5, higher early-season growth is observed at a few sites. We have explained this in the results section 3.1.1.1 of the manuscript.

    b. Yes, a variety of wheat cultivars are used across India. This is another major limitation of the model. However, for a model designed to simulate the regional impacts of land processes in an earth system, the complexity of including all cultivars of the crop will be overkill. For site scale simulation, it is essential to use crop cultivars-specific data.

4) 1.1.2 Yield – is it possible to separate grain yield from biomass growth, this would be an interesting distinction to make here for the site validation.

    a. We used yield in our analysis, which is the dry weight of the storage organ.

    b. We could not analyze above-ground biomass because the observational data on crop biomass is inconsistent in terms of units and measuring techniques.

    c. We are working on expanding the dataset to put together above-ground and below-ground biomass datasets so that they can be analyzed independently.

5) By drawing on the mean yield (t/ha) across sites, simulated vs. observed, you might be masking the model's strengths and weaknesses. A more transparent way of illustrating this performance metric would be beneficial. For example, the way you have illustrated LAI validation clearly shows that some sites are better simulated than others, which is normal and important to note.

a. Yes, that is a good point. The Taylor diagram masks the performance of the model at sites. Therefore, we plotted the MAB at each site overlayed on the India map (Figure 1 in this document) to show the regional performance of the model. We will replace Figure 4 (Taylor Diagram of LAI, yield, and growing season length) in the original manuscript with the figure shown below. We think this will help the reader understand the site-level model performance better.

b. The figure shown below clearly highlights which model performs better at each site. Ideally, we want CLM5_Mod2 to be the best across all sites. The best-performing model (having the lowest bias) will be the top-most marker at each site. For example, in Figure 1(b.2), CLM5_Mod1 is the best model (shown in the top marker in cyan color) at Kuthulia and Jabalpur, while CLM5_Mod2 is the best (shown in the top marker in blue color) at Hyderabad and Anantapur.

[Figure]

*Figure 1: Site-scale CLM performance against observations (1) Wheat; (2) Rice. Crop variables compared are (a) max. LAI during the growing season, (b) yield, and (c) growing season length. The three markers at each site location show the MAB of CLM5_Def (red color), CLM5_Mod1 (cyan color), and CLM5_Mod2 (blue color). The MAB ranges from 0 to 1. The contour on the map is the crop area per 0.5° grid cell.*

c. The addition of the new plot has led to an elaborate discussion of the CLM5 site-level performance in the results section. The text is added in several sections:

[revised manuscript text omitted]

6) In Figure 6, another neat way of illustrating this would be the spatial differences between obs and sim yields. – likewise for Figure 8 and GPP. Visually this could aid the interpretation of results.

a. Yes, this is a good suggestion. We will add the spatial difference figures for yield (Figure 3), GPP, and LAI (Figure 4), as shown below. Other spatial comparison plots of LH and SH in the supplementary will follow a similar style.

[Figure]

Figure 2: Yield estimates of (a) wheat and (b) rice by (1) EarthStat 2005, and (2-4) difference in yield between CLM5 (mean 2003-2007) versions and EarthStat data.

[Figure]

*Figure 3: Comparison of LAI simulated by CLM5 against MODIS data. The data shows the monthly LAI averaged over 2000-2014.*

7) Line 434: This is a very good point; in further work, it could be interesting to see whether it is possible to include the option of multiple rice harvests in one year (where agriculturally feasible) in CLM.

   a. CLM5 can have multiple harvests in one year for rice. The limiting factor in achieving this in the model is the availability of grid-level crop map data representing the rice cropping pattern across India. Limited crop maps capturing the multiple cropping of rice are available (e.g., Zhao et al. (2024) for the year 2019-20). The time-varying crop maps to conduct transient runs are missing.

8) With monthly time series of sensible and latent heat fluxes, you are essentially capturing how well the model captures seasonality. To dig deeper into how well these fluxes are simulated, it would be beneficial to uncover weekly, if not daily, fluxes.

   a. Yes, having a finer temporal resolution would give us a better understanding of the simulated sensible and latent heat fluxes. However, we did not do it for two reasons: 1) the objective of the study is to improve the seasonality of the crop growth in CLM5, and we wanted to emphasize the improvement in the fluxes at the seasonal scale. Hence, the monthly time series of fluxes are used. 2) The second reason for not producing the finer temporal resolution data is the lack of weekly or daily observational data on terrestrial fluxes to validate the result.

**Technical corrections:**

- Change "Site data used in validation" to "Site data used for validation" (in the supplement).
   a. Done
- Omit "the" in Line 37.
   a. Done
- In Line 125, "planting temperature" is repeated twice; delete one instance.

a. Done
- I would personally omit the code illustration in Lines 180 to 188 and keep the code in the supplementary material; a description in the main body suffices.
    a. Done
- In Line 496, when you mention that they are off by at least three months, it would be useful to specify whether the peak LAI by CLM5_Def is early or late.
    a. Added to the text

**Referee Comments III:**

**Summary**

The article focuses on improving the representation of two major Indian crops, spring wheat and rice, in the Community Land Model version 5.0 (CLM5) to enhance its accuracy in simulating crop phenology, yield, and associated land-atmosphere interactions. Using a newly created, comprehensive site-scale crop data set from India, the study calibrated and adjusted key parameters in the CLM5 crop module, such as sowing dates, growth parameters, and base temperature. The modified model versions (CLM5_Mod1 and CLM5_Mod2) demonstrated significant improvements in simulating crop growth, water and carbon fluxes, and irrigation patterns compared to the default CLM5 version. These modifications underscore the importance of region-specific parameters for global land models and provide a basis for better understanding land surface processes and their role in climate scenarios. The study's findings have implications for regional agricultural management and policy, as well as for enhancing climate modeling accuracy.

- We appreciate the thorough evaluation of the manuscript. Thank you for recognizing the value of the study and its importance in driving the Indian agroecosystems modeling studies.

The replies to the specific comments are as follows:

1) While the abstract states that it aims to improve the representation of Indian crops, it could be more specific earlier on by naming the two crops (spring wheat and rice). This would immediately inform the reader about the study's focus. Consider revising the sentence: "Our study aimed to improve the representation of these crops in CLM5" to "Our study aimed to improve the representation of spring wheat and rice in CLM5."

  a) Yes, the addition of spring and rice to the sentence would inform the readers better. The sentence in the abstract will be changed to "Our study aimed to improve the representation of spring wheat and rice in CLM5"

2) The abstract could briefly mention the broader implications of these improvements. For example, it could say, "These improvements can enhance the accuracy of land-atmosphere interaction studies and inform regional agricultural management and policy."

  a) This is a good suggestion, and the broader impact of these improvements will be added to the abstract. Please see the response to RC I: 6b.

3) While the introduction cites several relevant studies (e.g., Elliott et al., 2015; Lombardozzi et al., 2020), it could benefit from a few more recent references to highlight the current state of crop modeling. For example, "Recent studies provide valuable insights for enhancing the accuracy of simulating biogeophysical and biogeochemical processes..." could include more studies published after 2020 to strengthen this point.

  a) Thank you for pointing out the need to include more recent studies to strengthen the discussion on the current state of crop modeling. We agree that incorporating newer

research will enhance the introduction and provide a more comprehensive overview of recent advancements in crop modeling. Below, we have added citations to relevant studies published after 2020 and revised the text accordingly in line 42 of the manuscript.

"Recent studies provide valuable insights for enhancing the accuracy of simulating biogeophysical and biogeochemical processes at both regional and global scales in LSMs (Lobell et al., 2011; Osborne et al., 2015; Sheng et al., 2018; Lombardozzi et al., 2020; Boas et al., 2021; Ma et al., 2023)."

4) The introduction mentions that "CLM5 simulations of rice and wheat over the Indian subcontinent show large biases," but it could be clearer about what specific biases (e.g., underestimation of yield, incorrect phenology timing) the current study addresses. Adding a sentence such as, "Specifically, the model has been shown to inaccurately simulate the timing of planting and harvesting for spring wheat and rice, leading to incorrect estimates of carbon fluxes and water use," would provide a clearer problem statement.

   a) Thank you for pointing this out. Yes, specifying the kind and nature of bias would help the reader to understand the problem statement better. We are making the following changes to the manuscript in line 58 of the manuscript:

   "However, CLM5 simulations of rice and wheat over the Indian subcontinent show large biases in simulating annual crop yield (Lombardozzi et al., 2020). The major growing seasons of wheat and rice are the dry winter (rabi) season and wet monsoon (kharif) season, respectively, but CLM5 grows wheat and rice in the summer and rabi seasons, respectively."

5) The detailed methodological description would benefit from a flowchart or diagram summarizing the process, from data collection to model calibration and evaluation. For example, Figure 1 in the document effectively shows the sites used for calibration, but a flowchart could visually represent the steps outlined in Sections 2.1 to 2.3.1.

   a) This is a good recommendation. The following chart is prepared and will be added to the supplementary file.

[Figure]

6) The manuscript describes various parameter changes (e.g., base temperature, planting dates), but it could provide more justification for selecting these specific parameters for sensitivity analysis. For example, the section "Improvements in CLM5" states, "The base temperature and maximum GDD control the longevity of each phase in crop growth," but it does not explain why these were chosen over other potential parameters. A brief

explanation could be added, such as "These parameters were chosen based on their significant influence on phenological development stages in crops, as indicated by previous studies (cite studies)."

    a) The following text will be added to the text at line 136 of the manuscript:

    "The planting window, base temperature, GDD required for maturity, and grain fill parameters have a significant impact on crop growth and are considered when calibrating the crop module in CLM5 (Fisher et al., 2019; Cheng et al., 2020; Boas et al., 2021)."

7) While the Taylor diagrams (e.g., Figure 4) effectively show improvements in the model's performance, they could benefit from a brief explanation of how to interpret them. For example, "Higher correlation, lower RMS error, and smaller standard deviation characterize the most accurate CLM5 configuration, as seen in the closer proximity of CLM5_Mod2 markers to the observational reference point."

    a) See response to RC II- 5. The Taylor diagram in the original manuscript (Figure 4) is replaced by Figure 1 (in this document). However, we have another Taylor diagram in the manuscript (Figure 10). We will add the text to understand the model performance better near this figure explanation.

    b) We have included the way the Taylor diagram should be read in Section 2.4 from lines 258 to 266. The line at the end of the paragraph is "Higher correlation, lower RMS error, and smaller standard deviation characterize the most accurate CLM5 configuration". We hope that readers interpret the results accordingly. However, while interpreting the results from the Taylor Diagram for the first time in the manuscript at line 523, we will add the following statement: "Higher correlation, lower RMS error, and smaller standard deviation characterize the most accurate CLM5 configuration, as seen in the closer proximity of CLM5_Mod2 markers to the observational reference point."

8) The results section presents the remaining biases in yield and growing season length (e.g., "The growing season length simulated by CLM5_Def is very low with a mean growing season of just 69 days, compared to 129 days in observations"), but it could discuss potential reasons for these biases in more detail. For example, it could mention model assumptions, data limitations, or unaccounted-for environmental factors that could contribute to these discrepancies.

    a) The discrepancies in the growing season length of wheat crops are due to the growth in the warm season and low base temperature. This reason is added to the text to enable the reader to understand the cause of the high bias in line 353 of the manuscript: "Incorrect growing season and a low $T_{base}$ for wheat have led to the simulation of very low growing season length in the CLM5_Def."

9) The manuscript mentions, "Such improved land models will be a great asset in investigating global and regional-scale land-atmosphere interactions and developing future climate scenarios," but it could expand on specific applications. For instance, how could this model be used to inform irrigation management practices or forecast agricultural productivity under different climate scenarios?

    a) The following text is added to the manuscript at the line 616:

"Models that can simulate regional crop and land processes accurately will be able to predict the future water demand of the crops and if enough water sources are available to meet the needs. They can also help in providing estimates of productivity and net carbon capture abilities of agroecosystems in future climate."

10) The discussion would benefit from a dedicated section on limitations and future directions. For example, the text could state, "While the modified models showed significant improvements, there are still biases that could be addressed by incorporating more diverse site data or accounting for multi-cropping practices in the model," and suggest specific future research that could address these limitations.

   a) This is a good suggestion. We have expanded on the future scope of work available and the major shortcomings of the work in the paragraph at line 586 in the manuscript. However, it is important to specify the direction in which future work should head. Therefore, we are adding the following text to the manuscript in line 594:

"The major drawback of the study is not considering the multiple cropping of rice followed in major parts of India. Although the harvested area of rice grown in seasons other than those considered in this study is very low, it is important to include it as rice growth has a significant impact on terrestrial fluxes (Oo et al., 2023)."

11) The conclusion could be strengthened by including a call for further studies or actions, such as "Future work could focus on extending this modeling approach to other major crops in India or integrating socio-economic factors to better inform policy-making."

   a) See the response to RC III: 10.

12) Consider adding more comparative visuals that summarize the improvements across different metrics and model versions. For instance, a bar chart comparing MAB, RMSE, and Pearson's r values for CLM5_Def, CLM5_Mod1, and CLM5_Mod2 could provide a quick visual reference for readers.

   a) Taylor's diagram is a good visual comparison. However, we are adding the following figure to condense the information in Table 3 (to the supplement), and we feel this figure helps in conveying the improvements made in this study.

[Figure]

*Figure 4: Bar plot of crop parameters (a) max. LAI, (b) Yield, and (c) growing season length and the evaluation metrics used in this study (1) Mean over all sites, (2) Mean Absolute Bias (MAB), (3) Root Mean Square Error (RMSE), (4) Pearsons*

*r, and (5) Kling-Gupta Efficiency (KGE). The panel d.2 shows the overall bias of individual CLM5 model in simulating the crop parameters.*

- This figure will be added as a supplementary to the manuscript.

13) Include more recent references (post-2020) to ensure the manuscript reflects the latest advancements in the field. For example, search for recent studies on crop modeling in LSMs that may have incorporated new methodologies or datasets.

    a) See the response to RC III-3.

14) The manuscript is generally well-written with a clear technical style suitable for the target audience. However, some sections could benefit from simplified language to increase accessibility for readers from diverse scientific backgrounds.

    The manuscript follows a logical flow, but the Methods and Discussion sections could be further refined for clarity and depth.

    a) Additions in text and a few figures to facilitate the reader are made to the methods and discussion section (See the responses to RC III:  5, 6, 8, 9, and 10; and RC II: 5)

15) Provide more justification for the choice of specific parameters in the sensitivity analysis and model modifications.

    a) See the response to RC III: 6.

16) Include a discussion on the potential policy and practical implications of the findings and suggest specific areas for future research.

    a) See the response to RC III: 9 and 10.

17) Consider adding more comparative figures and diagrams to succinctly showcase the differences between model versions and their performance improvements.

    - See response to RC III: 12 and RC II: 5.

---

## Author Response (AR2)

**Improving the representation of major Indian crops in the Community Land Model version 5.0 (CLM5) using site-scale crop data**

K. Narender Reddy, Somnath Baidya Roy, Sam S. Rabin, Danica L. Lombardozzi, Gudimetla Venkateswara Varma, Ruchira Biswas, and Devavat Chiru Naik

We thank the referees for their suggestion to further improve the quality of the manuscript and the editor for allowing us to respond to the referees' comments.

Below, we provide a point-by-point response to the referees' comments. The comments provided by the referee are in red color, our responses are in black color, and the proposed changes to the text are in green color.

Referee I:

1) By way of final recommendations or suggestions to further improve the manuscript, in the methods section, a visual representation of the workflow would be helpful to readers - authors should include a flowchart summarizing methodological steps, from data collection to model evaluation, to improve accessibility. Finally in the discussion section, explicitly state the limitations, such as assumptions about single-season rice crops, and discuss how they might affect the findings. Other than that, the manuscript is primed for publication.

a) A visual representation of the methodology is added to the supplementary material in the last revision. An extra text is added to the manuscript to direct the readers to this flowchart in the line 102 of the revised manuscript: "The overall methodology and steps followed in this study are depicted as a flowchart (Figure S1) and explained in detail in the following sections."

b) To highlight the major drawbacks of the study and its consequences, the following text is added to the revised manuscript at the line 638: "Although the harvested area of rice grown in rabi and summer seasons is very low (Biemans et al., 2016), it is important to include the rice growth in these seasons in LSMs. This will significantly impact the terrestrial fluxes at the local scale (Oo et al., 2023). The lower LH simulated by the CLM5 models during the rabi and summer season (November to June) compared to FLUXCOM data (Figure S12(a)) might be due to growing rice in kharif season only. However, because of the small areal coverage of rabi and summer rice, their impact on large scale fluxes and weather/climate is likely to be small. This study did not consider other major crops, such as maize, soybean, and pulses, which cover substantial harvesting areas. Future studies should focus on improving the representation of these crops in CLM5 for a comprehensive study of climate impacts on Indian agroecosystems."

In addition to the above mentioned changes, the manuscript is thoroughly checked for typos, grammatical errors and a few sentence corrections (highlighted in tracked changes file). For example, usage of "spring wheat" and "wheat" interchangeably in the manuscript might confuse the reader and therefore, we consistently used "wheat" in the manuscript.